# Energy-Based Sliced Wasserstein Distance

**Khai Nguyen**
Department of Statistics and Data Sciences
The University of Texas at Austin
Austin, TX 78712
khainb@utexas.edu

**Nhat Ho**
Department of Statistics and Data Sciences
The University of Texas at Austin
Austin, TX 78712
minhnhat@utexas.edu

## Abstract

The sliced Wasserstein (SW) distance has been widely recognized as a statistically effective and computationally efficient metric between two probability measures. A key component of the SW distance is the slicing distribution. There are two existing approaches for choosing this distribution. The first approach is using a fixed prior distribution. The second approach is optimizing for the best distribution which belongs to a parametric family of distributions and can maximize the expected distance. However, both approaches have their limitations. A fixed prior distribution is non-informative in terms of highlighting projecting directions that can discriminate two general probability measures. Doing optimization for the best distribution is often expensive and unstable. Moreover, designing the parametric family of the candidate distribution could be easily misspecified. To address the issues, we propose to design the slicing distribution as an energy-based distribution that is parameter-free and has the density proportional to an energy function of the projected one-dimensional Wasserstein distance. We then derive a novel sliced Wasserstein variant, *energy-based sliced Waserstein* (EBSW) distance, and investigate its topological, statistical, and computational properties via importance sampling, sampling importance resampling, and Markov Chain methods. Finally, we conduct experiments on point-cloud gradient flow, color transfer, and point-cloud reconstruction to show the favorable performance of the EBSW[1].

## 1 Introduction

The sliced Wasserstein [2] (SW) distance is a sliced probability metric that is derived from the Wasserstein distance [5, 36] as the base metric. Utilizing the closed-form solution of optimal transport on one-dimension [36], the SW distance can be computed very efficiently at the time complexity of $\mathcal{O}(n \log n)$ and the space complexity of $\mathcal{O}(n)$ when dealing with two probability measures that have at most $n$ supports. Moreover, the sample complexity of the SW is only $\mathcal{O}(n^{-1/2})$ [27, 34] which indicates that it does not suffer from the curse of dimensionality in statistical inference. Therefore, the SW distance has been applied to various domains of applications including point-cloud applications e.g., reconstruction, registration, generation, and upsampling [31, 41], generative models [8], domain adaptation [22], clustering [20], gradient flows [24, 1], approximate Bayesian computation [26], variational inference [47], and many other tasks.

The SW distance can be defined as the expectation of the one-dimensional Wasserstein distance between two projected probability measures. The randomness comes from a random projecting direction which is used to project two original probability measures to one dimension. The probability distribution of the random projecting direction is referred to as the slicing distribution. Therefore, a central task that decides the effectiveness of the SW in downstream applications is designing

---

[1]Code for this paper is published at https://github.com/khainb/EBSW.

37th Conference on Neural Information Processing Systems (NeurIPS 2023).

slicing distribution. The conventional sliced Wasserstein distance [2] simply takes the uniform distribution over the unit-hypersphere as the slicing distribution. Despite being easy to sample from, the uniform distribution is not able to differentiate between informative and non-informative projecting distributions in terms of discriminating two interested probability measures through projection [7]. To avoid a flat prior distribution like the uniform distribution, a different approach tries to find the best slicing distribution that can maximize the expectation. This distribution is found inside a parametric family of distribution over the unit-hypersphere [30, 32]. However, searching for the best slicing distribution often requires an iterative procedure which is often computationally expensive and unstable. Moreover, choosing the family for the slicing distribution is challenging since the number of distributions over the unit-hypersphere is limited. Widely used and explicit spherical distributions such as von Mises-Fisher distribution [17, 32] might be misspecified while implicit distributions [30] are expensive, hard to adapt to downstream applications and uninterpretable.

In this paper, we aim to develop new choices of slicing distributions that are both discriminative in comparing two given probability measures and do not require optimization. Motivated by energy-based models [25, 16], we model the slicing distribution by an unnormalized density function which gives a higher density for a more discriminative projecting direction. To induce that property, the density function at a projecting direction is designed to be proportional to the value of the one-dimensional Wasserstein distance between the two corresponding projected probability measures.

**Contribution.** In summary, our contributions are three-fold:

1. We propose a new class of slicing distribution, named *energy-based slicing* distribution, which has the density function proportional to the value of the projected one-dimensional Wasserstein distance. We further control the flexibility of the slicing distribution by applying an energy function e.g., the polynomial function, and the exponential function, to the projected Wasserstein value. By using the energy-based slicing distribution, we derive a novel metric on the space of probability measures, named *energy-based sliced Wasserstein* (EBSW) distance.

2. We derive theoretical properties of the proposed EBSW including topological properties, statistical properties, and computational properties. For topological properties, we first prove that the EBSW is a valid metric on the space of probability measures. After that, we show that the weak convergence of probability measures is equivalent to the convergence of probability measures under the EBSW distance. Moreover, we develop the connection of the EBSW to existing sliced Wasserstein variants and the Wasserstein distance. We show that the EBSW is the first non-optimization variant that is an upper bound of the sliced Wasserstein distance. For the statistical properties, we first derive the sample complexity of the EBSW which indicates that it does not suffer from the curse of dimensionality. For computational properties, we propose importance sampling, sampling importance resampling, and Markov Chain Monte Carlo methods to derive empirical estimations of the EBSW. Moreover, we discuss the time complexities and memory complexities of the corresponding estimations. Finally, we discuss the statistical properties of estimations.

3. We apply the EBSW to various tasks including gradient flows, color transfer, and point-cloud applications. According to the experimental result, the EBSW performs better than existing *projection-selection* sliced Wasserstein variants including the conventional sliced Wasserstein [2] (SW), max sliced Wasserstein [7] (Max-SW), and distributional sliced Wasserstein (DSW) [30, 32]. More importantly, the importance sampling estimation of the EBSW is as efficient and easy to implement as the conventional SW, i.e., its implementation can be obtained by adding one to two lines of code.

**Organization.** The remainder of the paper is organized as follows. We first review the background on the sliced Wasserstein distance and its projection-selection variants in Section 2. We then define the energy-based sliced Wasserstein distance, derive their theoretical properties, and discuss its computational methods in Section 3. Section 4 contains experiments on gradient flows, color transfer, and point-cloud applications. We conclude the paper in Section 5. Finally, we defer the proofs of key results, and additional materials in the Appendices.

**Notations.** For any $d \geq 2$, we denote $\mathbb{S}^{d-1} := \{\theta \in \mathbb{R}^d \mid ||\theta||_2^2 = 1\}$ and $\mathcal{U}(\mathbb{S}^{d-1})$ as the unit hyper-sphere and its corresponding uniform distribution . We denote $\mathcal{P}(\mathcal{X})$ as the set of all probability measures on the set $\mathcal{X}$. For $p \geq 1$, $\mathcal{P}_p(\mathcal{X})$ is the set of all probability measures on the set $\mathcal{X}$ that have finite $p$-moments. For any two sequences $a_n$ and $b_n$, the notation $a_n = \mathcal{O}(b_n)$ means that $a_n \leq C b_n$ for all $n \geq 1$, where $C$ is some universal constant. We denote $\theta \sharp \mu$ is the push-forward measures of $\mu$

through the function $f : \mathbb{R}^d \to \mathbb{R}$ that is $f(x) = \theta^\top x$. For a vector $X \in \mathbb{R}^{dm}$, $X := (x_1, \ldots, x_m)$, $P_X$ denotes the empirical measures $\frac{1}{m} \sum_{i=1}^{m} \delta_{x_i}$.

## 2 Background

In this section, we first review the sliced Wasserstein distance and its projection-selection variants including max sliced Wasserstein distance and distributional sliced Wasserstein distance.

**Sliced Wasserstein.** The definition of sliced Wasserstein (SW) distance [2] between two probability measures $\mu \in \mathcal{P}_p(\mathbb{R}^d)$ and $\nu \in \mathcal{P}_p(\mathbb{R}^d)$ is:

$$\mathrm{SW}_p(\mu, \nu) = \left( \mathbb{E}_{\theta \sim \mathcal{U}(\mathbb{S}^{d-1})} [\mathrm{W}_p^p(\theta \sharp \mu, \theta \sharp \nu)] \right)^{\frac{1}{p}}, \tag{1}$$

where the Wasserstein distance has a closed form which is $\mathrm{W}_p^p(\theta \sharp \mu, \theta \sharp \nu) = \int_0^1 |F_{\theta \sharp \mu}^{-1}(z) - F_{\theta \sharp \nu}^{-1}(z)|^p dz$ where $F_{\theta \sharp \mu}$ and $F_{\theta \sharp \nu}$ are the cumulative distribution function (CDF) of $\theta \sharp \mu$ and $\theta \sharp \nu$ respectively. However, the expectation in the definition of the SW distance is intractable to compute. Therefore, the Monte Carlo scheme is employed to approximate the value:

$$\widehat{\mathrm{SW}}_p(\mu, \nu; L) = \left( \frac{1}{L} \sum_{l=1}^{L} \mathrm{W}_p^p(\theta_l \sharp \mu, \theta_l \sharp \nu) \right)^{\frac{1}{p}}, \tag{2}$$

where $\theta_1, \ldots, \theta_L \overset{i.i.d}{\sim} \mathcal{U}(\mathbb{S}^{d-1})$ and are referred to as projecting directions. The pushforward measures $\theta_1 \sharp \mu, \ldots, \theta_L \sharp \mu$ are called projections of $\mu$ (similar to $\nu$). The number of Monte Carlo samples $L$ is often referred to as the number of projections. When $\mu$ and $\nu$ are discrete measures that have at most $n$ supports, the time complexity and memory complexity of the SW are $\mathcal{O}(Ln \log n)$ and $\mathcal{O}(L(d + n))$ respectively. It is worth noting that $\widehat{\mathrm{SW}}_p^p(\mu, \nu; L)$ is an unbiased estimation of $\mathrm{SW}_p^p(\mu, \nu)$, however, $\widehat{\mathrm{SW}}_p(\mu, \nu; L)$ is only asymptotically unbiased estimation of $\mathrm{SW}_p(\mu, \nu)$. Namely, we have $\widehat{\mathrm{SW}}_p(\mu, \nu; L) \to \mathrm{SW}_p(\mu, \nu)$ when $L \to \infty$ (law of large numbers).

**Distributional sliced Wasserstein.** As discussed, using the uniform distribution over projecting directions is not suitable for two general probability measures. A natural extension is to replace the uniform distribution with a "better" distribution on the unit-hypersphere. Distributional sliced Wasserstein [30] suggests searching this distribution in a parametric family of distributions by maximizing the expected distance. The definition of distributional sliced Wasserstein (DSW) distance [30] between two probability measures $\mu \in \mathcal{P}_p(\mathbb{R}^d)$ and $\nu \in \mathcal{P}_p(\mathbb{R}^d)$ is:

$$\mathrm{DSW}_p(\mu, \nu) = \max_{\psi \in \Psi} \left( \mathbb{E}_{\theta \sim \sigma_\psi(\theta))} [\mathrm{W}_p^p(\theta \sharp \mu, \theta \sharp \nu)] \right)^{\frac{1}{p}}, \tag{3}$$

where $\sigma_\psi(\theta) \in \mathcal{P}(\mathbb{S}^{d-1})$, e.g., von Mises-Fisher [17] distribution with unknown location parameter $\sigma_\psi(\theta) := \mathrm{vMF}(\theta|\epsilon, \kappa)$, $\psi = \epsilon$ [32]. After using $T \geq 1$ (projected) stochastic (sub)-gradient ascent iterations to obtain an estimation of the parameter $\hat{\psi}_T$, Monte Carlo samples $\theta_1, \ldots, \theta_L \overset{i.i.d}{\sim} \sigma_{\hat{\psi}_T}(\theta)$ are used to approximate the value of the DSW. Interestingly, the metricity DSW holds for non-optimal $\hat{\psi}_T$ as long as $\sigma_{\hat{\psi}_T}(\theta)$ are continuous on $\mathbb{S}^{d-1}$ e.g., vMF with $\kappa < \infty$ [31]. In addition, the unbiasedness property of the DSW is the same as the SW, namely, when $L \to \infty$, the empirical estimation of the DSW converges to the true value. The time complexity and space complexity of the DSW are $\mathcal{O}(LTn \log n)$ and $\mathcal{O}(L(d + n))$ in turn without counting the complexities of sampling from $\sigma_{\hat{\psi}_T}(\theta)$. We refer to Appendix B.1 for more details, e.g., equations, algorithms, and discussion. PAC-Bayesian generalization bounds for DSW are investigated in [35].

**Max sliced Wasserstein.** By letting the concentration parameter $\kappa \to \infty$, the vMF distribution degenerates to the Dirac distribution $\mathrm{vMF}(\theta|\epsilon, \kappa) \to \delta_\epsilon$, we obtain the max sliced Wasserstein distance [7]. The definition of max sliced Wasserstein (Max-SW) distance between two probability measures $\mu \in \mathcal{P}_p(\mathbb{R}^d)$ and $\nu \in \mathcal{P}_p(\mathbb{R}^d)$ is:

$$\mathrm{Max\text{-}SW}_p(\mu, \nu) = \max_{\theta \in \mathbb{S}^{d-1}} \mathrm{W}_p(\theta \sharp \mu, \theta \sharp \nu). \tag{4}$$

Similar to the DSW, the Max-SW is often computed by using $T \geq 1$ iterations of (projected) (sub)-gradient ascent to obtain an estimation of the "max" projecting direction $\hat{\theta}_T$. After that, the

estimated value of the Max-SW is $W_p(\hat{\theta}_T \sharp \mu, \hat{\theta}_T \sharp \nu)$. The time complexity and space complexity of the Max-SW are $\mathcal{O}(Tn \log n)$ and $\mathcal{O}(d + n)$. It is worth noting that the Max-SW is only a metric at the global optimum $\theta^\star$, hence, we cannot guarantee the metricity of the Max-SW due to the non-convex optimization [34] problem even when $T \to \infty$. Therefore, the performance of Max-SW is often unstable in practice [28]. We refer the reader to Appendix B.1 for more details e.g., equations, algorithms, and discussions about the Max-SW.

## 3 Energy-Based Sliced Wasserstein Distance

From the background, we observe that using a fixed slicing distribution e.g., in the SW, is computationally efficient but might be not effective. In contrast, using optimization-based slicing distributions is computationally expensive e.g., in the DSW, and is unstable e.g., in the Max-SW. Therefore, we address previous issues by introducing a novel sliced Wasserstein variant that uses optimization-free slicing distribution which can highlight the difference between two comparing probability measures.

### 3.1 Energy-Based Slicing Distribution

We first start with the key contribution which is the energy-based slicing distribution.

**Definition 1.** *For any $p \geq 1$, dimension $d \geq 1$, an energy function $f : [0, \infty) \to \Theta \subset (0, \infty)$ and two probability measures $\mu \in \mathcal{P}_p(\mathbb{R}^d)$ and $\nu \in \mathcal{P}_p(\mathbb{R}^d)$, the energy-based slicing distribution $\sigma_{\mu,\nu}(\theta)$ supported on $\mathbb{S}^{d-1}$ is defined as follow:*

$$\sigma_{\mu,\nu}(\theta; f, p) \propto f(W_p^p(\theta \sharp \mu, \theta \sharp \nu)) := \frac{f(W_p^p(\theta \sharp \mu, \theta \sharp \nu))}{\int_{\mathbb{S}^{d-1}} f(W_p^p(\theta \sharp \mu, \theta \sharp \nu)) d\theta}, \tag{5}$$

*where the image of $f$ is in the open interval $(0, \infty)$ is for making $\sigma_{\mu,\nu}(\theta)$ continuous on $\mathbb{S}^{d-1}$*

In contrast to the approach of the DSW which creates the dependence between the slicing distribution and two input probability measures via optimization, the energy-based slicing distribution obtains the dependence by exploiting the value of the projected Wasserstein distance between two input probability measures at each support.

**Monotonically increasing energy functions.** Similar to previous works, we again assume that *"A higher value of projected Wasserstein distance, a better projecting direction"*. Therefore, it is natural to use a monotonically increasing function for the energy function $f$. We consider the following two functions: the exponential function: $f_e(x) = e^x$, and the shifted polynomial function: $f_q(x) = x^q + \varepsilon$ with $q, \varepsilon > 0$. The shifted constant $\varepsilon$ helps to avoid the slicing distribution undefined when two input measures are equal. In a greater detail, when $\mu = \nu$, we have $W_p^p(\theta \sharp \mu, \theta \sharp \nu) = 0$ for all $\theta \in \mathbb{S}^{d-1}$ due to the identity property of the Wasserstein distance. Hence, $\sigma_{\mu,\nu}(\theta; f, p) \propto 0$ for $\theta \in \mathbb{S}^{d-1}$ and $f(x) = x^q$ ($q > 0$). Therefore, the slicing distribution $\sigma_{\mu,\nu}(\theta; f, p)$ is undefined due to an invalid density function. In practice, it is able to set $\varepsilon = 0$ since we rarely deal with two coinciding measures.

**Other energy functions.** We can choose any positive function for energy function $f$ and it will result in a valid slicing distribution. However, it is necessary to come up with an assumption for the choice of the function. Since there is no existing other assumption for the importance of projecting direction, we will leave the investigation of non-increasing energy function $f$ to future works.

**Example 1.** *Let $\mu = \mathcal{N}(\mathbf{m}, v_1^2 \mathbf{I})$ and $\nu = \mathcal{N}(\mathbf{m}, v_2^2 \mathbf{I})$ are two location-scale Gaussian distributions with the same means, we have their projections are $\theta \sharp \mu = \mathcal{N}(\theta^\top \mathbf{m}, v_1^2)$ and $\theta \sharp \nu = \mathcal{N}(\theta^\top \mathbf{m}, v_2^2)$. Based on the closed form of the Wasserstein distance between two Gaussians [10], we have $W_2^2(\theta \sharp \mu, \theta \sharp \nu) = (v_1 - v_2)^2$ for all $\theta \in \mathbb{S}^{d-1}$ which leads to $\sigma_{\mu,\nu}(\theta; f, p) = \mathcal{U}(\mathbb{S}^{d-1})$ for definitions of energy function $f$ in Definition 1.*

Example 1 gives a special case where we can have the closed form of the slicing function.

**Applications to other sliced probability metrics and mutual information.** In this paper, we focus on comparing choices of slicing distribution in the basic form of the SW distance. The proposed energy-based slicing distribution can be adapted to other variants of the SW that are not about designing slicing distribution e.g., non-linear projecting [19], orthogonal projecting directions [38], and so on. Moreover, the energy-based slicing approach can be applied to other sliced probability metrics e.g., sliced score matching [42], and sliced mutual information [13].

## 3.2 Definitions, Topological, and Statistical Properties of Energy Based Sliced Wasserstein

With the definition of energy-based slicing distribution in Definition 1, we now are able to define the energy-based sliced Wasserstein (EBSW) distance.

**Definition 2.** *For any $p \geq 1$, dimension $d \geq 1$, two probability measures $\mu \in \mathcal{P}_p(\mathbb{R}^d)$ and $\nu \in \mathcal{P}_p(\mathbb{R}^d)$, the energy function $f : [0, \infty) \to (0, \infty)$, and the energy-based slicing distribution $\sigma_{\mu,\nu}(\theta; f, p)$, the energy-based sliced Wasserstein (EBSW) distance is defined as follows:*

$$EBSW_p(\mu, \nu; f) = \left( \mathbb{E}_{\theta \sim \sigma_{\mu,\nu}(\theta; f, p)} \left[ W_p^p(\theta \sharp \mu, \theta \sharp \nu) \right] \right)^{\frac{1}{p}}. \tag{6}$$

We now derive some theoretical properties of the EBSW distance.

**Topological Properties.** We first investigate the metricity of the EBSW distance.

**Theorem 1.** *For any $p \geq 1$, energy-function $f$, the energy-based sliced Wasserstein $EBSW_p(\cdot, \cdot; f)$ is a semi-metric in the probability space on $\mathbb{R}^d$, namely EBSW satisfies non-negativity, symmetry, and identity of indiscernibles.*

The proof of Theorem 1 in given in Appendix A.1. Next, we establish the connections among the EBSW, the SW, the Max-SW, and the Wasserstein.

**Proposition 1.** *(a) For any $p \geq 1$ and increasing energy function $f$, we find that*

$$SW_p(\mu, \nu) \leq EBSW_p(\mu, \nu; f).$$

*The equality holds when $f(x) = c$ for some positive constant $c$ for all $x \in [0, \infty)$.*

*(b) For any $p \geq 1$ and energy function $f$, we have*

$$EBSW_p(\mu, \nu; f) \leq \text{Max-SW}_p(\mu, \nu) \leq W_p(\mu, \nu).$$

Proof of Proposition 1 is in Appendix A.2. The results of Proposition 1 indicate that for increasing energy function $f$, the EBSW is lower bounded by the SW while it is upper bounded by the Max-SW. It is worth noting that the EBSW is the first variant that changes the slicing distribution while still being an upper bound of the SW.

**Theorem 2.** *For any $p \geq 1$ and energy function $\bar{f}$, the convergence of probability measures under the energy-based sliced Wasserstein distance $EBSW_p(\cdot, \cdot; \bar{f})$ implies weak convergence of probability measures and vice versa.*

Theorem 2 implies that for any sequence of probability measures $(\mu_k)_{k \in \mathbb{N}}$ and $\mu$ in $\mathcal{P}_p(\mathbb{R}^d)$, $\lim_{k \to +\infty} EBSW_p(\mu_k, \mu; \bar{f}) = 0$ if and only if for any continuous and bounded function $f : \mathbb{R}^d \to \mathbb{R}$, $\lim_{k \to +\infty} \int f \, d\mu_k = \int f \, d\mu$. The proof of Theorem 2 is in Appendix A.3.

**Statistical Properties.** From Proposition 1, we derive the sample complexity of the EBSW.

**Proposition 2.** *Let $X_1, X_2, \ldots, X_n$ be i.i.d. samples from the probability measure $\mu$ being supported on compact set of $\mathbb{R}^d$. We denote the empirical measure $\mu_n = \frac{1}{n} \sum_{i=1}^{n} \delta_{X_i}$. Then, for any $p \geq 1$ and energy function $f$, there exists a universal constant $C > 0$ such that*

$$\mathbb{E}[EBSW_p(\mu_n, \mu; f)] \leq C\sqrt{(d+1) \log n/n},$$

*where the outer expectation is taken with respect to the data $X_1, X_2, \ldots, X_n$.*

The proof of Proposition 2 is given in Appendix A.4. From this proposition, we can say that the EBSW does not suffer from the curse of dimensionality. We will discuss other statistical properties of approximating the EBSW in the next section.

### 3.3 Computational Methods and Computational Properties

Calculating the expectation with respect to the slicing distribution $\sigma_{\mu,\nu}(\theta; f, p)$ is intractable. Therefore, we propose some Monte Carlo estimation methods to approximate the value of EBSW.

### 3.3.1 Importance Sampling

The most simple and computationally efficient method that can be used is importance sampling (IS) [18]. The idea is to utilize an efficient-sampling proposal distribution $\sigma_0(\theta) \in \mathcal{P}(\mathbb{S}^{d-1})$ to provide Monte Carlo samples. After that, we use the density ratio between the original slicing distribution and the proposal distribution to weight samples. We can rewrite the EBSW distance as:

$$\text{EBSW}_p(\mu, \nu; f) = \left( \frac{\mathbb{E}_{\theta \sim \sigma_0(\theta)} \left[ \text{W}_p^p(\theta \sharp \mu, \theta \sharp \nu) w_{\mu,\nu,\sigma_0,f,p}(\theta) \right]}{\mathbb{E}_{\theta \sim \sigma_0(\theta)} \left[ w_{\mu,\nu,\sigma_0,f,p}(\theta) \right]} \right)^{\frac{1}{p}}, \tag{7}$$

where $\sigma_0(\theta) \in \mathcal{P}(\mathbb{S}^{d-1})$ is the proposal distribution, and:

$$w_{\mu,\nu,\sigma_0,f,p}(\theta) = \frac{f(\text{W}_p^p(\theta \sharp \mu, \theta \sharp \nu))}{\sigma_0(\theta)}$$

is the importance weighted function. The detailed derivation is given in Appendix B.2. Let $\theta_1, \ldots, \theta_L$ be i.i.d samples from $\sigma_0(\theta)$, the importance sampling estimator of the EBSW (IS-EBSW) is:

$$\widehat{\text{IS-EBSW}}_p(\mu, \nu; f, L) = \left( \sum_{l=1}^{L} \left[ \text{W}_p^p(\theta_l \sharp \mu, \theta_l \sharp \nu) \hat{w}_{\mu,\nu,\sigma_0,f,p}(\theta_l) \right] \right)^{\frac{1}{p}}, \tag{8}$$

where $\hat{w}_{\mu,\nu,\sigma_0,f,p}(\theta_l) = \frac{w_{\mu,\nu,\sigma_0,f,p}(\theta_l)}{\sum_{l'=1}^{L} w_{\mu,\nu,\sigma_0,f,p}(\theta_{l'})}$ is the normalized importance weights. When $\sigma_0(\theta) = \mathcal{U}(\mathbb{S}^{d-1}) = \frac{\Gamma(d/2)}{2\pi^{d/2}}$ (a constant of $\theta$), we can replace $w_{\mu,\nu,\sigma_0}(\theta_l)$ with $f(\text{W}_p^p(\theta_l \sharp \mu, \theta_l \sharp \nu))$. When we choose the energy function $f(x) = e^x$, computing the normalized importance weights

$$\hat{w}_{\mu,\nu,\sigma_0,f,p}(\theta_l) = \frac{w_{\mu,\nu,\sigma_0,f,p}(\theta_l)}{\sum_{l'=1}^{L} w_{\mu,\nu,\sigma_0,f}(\theta_{l'})}$$

is equivalent to computing the Softmax function.

**Computational algorithms and complexities.** The computational algorithm of IS-EBSW can be derived from the algorithm of the SW distance by adding only one to two lines of code for computing the importance weights. For a better comparison, we give algorithms for computing the SW distance and the EBSW distance in Algorithm 1 and Algorithm 4 in Appendix B.1 and Appendix B.2 respectively. When $\mu$ and $\nu$ are two discrete measures that have at most $n$ supports, the time complexity and the space complexity of the IS-EBSW distance are $\mathcal{O}(Ln \log n + Lnd)$ and $\mathcal{O}(L(n+d))$ which are the same as the SW.

**Unbiasedness.** The IS approximation is asymptotically unbiased for $\text{EBSW}_p^p(\mu, \nu; f)$. However, having a biased estimation is not severe since the unbiasedness cannot be preserved after taking the $p$-rooth ($p > 1$) like in the case of the SW distance. Therefore, an unbiased estimation of $\text{EBSW}_p^p(\mu, \nu; f)$ is not very vital. Moreover, we can show that $\widehat{\text{IS-EBSW}}_p^p(\mu, \nu; f, L)$ is an unbiased estimation of the power $p$ of a valid distance. We refer the reader to Appendix B.2 for the detailed definition and properties of the distance. From this insight, it is safe to use the IS-EBSW in practice.

**Gradient Estimation.** In statistical inference, we might want to estimate the gradient $\nabla_\phi \text{EBSW}_p(\mu_\phi, \nu; f)$ for doing minimum distance estimator [45]. Therefore, we derive the gradient estimator of the EBSW with importance sampling in Appendix B.2.

### 3.3.2 Sampling Importance Resampling and Markov Chain Monte Carlo

The second approach is to somehow sample from the slicing distribution $\sigma_{\mu,\nu}(\theta; f, p)$. For example, when we have $\theta_1, \ldots, \theta_L$ are approximately distributed following $\sigma_{\mu,\nu}(\theta; f, p)$, we can take $\left( \frac{1}{L} \sum_{l=1}^{L} \text{W}_p^p(\theta_l \sharp \mu, \theta_l \sharp \nu) \right)^{\frac{1}{p}}$ as the approximated value of the EBSW. Here, we consider two famous approaches in statistics: Sampling Importance Resampling [14] (SIR) and Markov Chain Monte Carlo (MCMC). For MCMC, we utilize two variants of the Metropolis-Hasting algorithm: independent Metropolis-Hasting (IMH) and random walk Metropolis-Hasting (RMH).

**Sampling Importance Resampling.** Similar to importance sampling in Section 3.3.1, the SIR uses a proposal distribution $\sigma_0(\theta)$ to obtain $L$ samples $\theta_1', \ldots, \theta_L'$ and the corresponding normalized

importance weights:
$$\hat{w}_{\mu,\nu,\sigma_0,f,p}(\theta_l') = \frac{w_{\mu,\nu,\sigma_0,f,p}(\theta_l')}{\sum_{i=1}^{L} w_{\mu,\nu,\sigma_0,f,p}(\theta_i')}.$$

After that, the SIR creates the resampling distribution which is a Categorical distribution $\hat{q}(\theta) = \sum_{l=1}^{L} \hat{w}_{\mu,\nu,\sigma_0,f,p}(\theta_l)\delta_{\theta_{l'}}$. Finally, the SIR draws $L$ samples $\theta_1, \ldots, \theta_L \overset{i.i.d}{\sim} \hat{q}(\theta)$. We denote the SIR estimation of the EBSW distance as SIR-EBSW.

**Markov Chain Monte Carlo.** MCMC creates a Markov chain that has the stationary distribution as the target distribution. The most famous way to construct such a Markov chain is through the Metropolis-Hastings algorithm. Let the starting sample follow a prior distribution $\theta_1 \sim \sigma_0(\theta) \in \mathcal{P}(\mathbb{S}^{d-1})$, a transition distribution $\sigma_t(\theta_t|\theta_{t-1}) \in \mathcal{P}(\mathbb{S}^{d-1})$ for any timestep $t > 1$ is used to sample a candidate $\theta_t'$. After that, the new sample $\theta_t$ is set to $\theta_t'$ with the probability $\alpha$ and is set to $\theta_{t-1}$ with the probability $1 - \alpha$ with

$$\alpha = \min\left(1, \frac{\sigma_{\mu,\nu}(\theta_t'; f)}{\sigma_{\mu,\nu}(\theta_{t-1}; f)} \frac{\sigma_t(\theta_{t-1}|\theta_t')}{\sigma_t(\theta_t'|\theta_{t-1})}\right) = \min\left(1, \frac{f(\mathrm{W}_p^p(\theta_t'\sharp\mu, \theta_t'\sharp\nu)))}{f(\mathrm{W}_p^p(\theta_{t-1}\sharp\mu, \theta_{t-1}\sharp\nu)))} \frac{\sigma_t(\theta_{t-1}|\theta_t')}{\sigma_t(\theta_t'|\theta_{t-1})}\right).$$

In theory, $T$ should be large enough to help the Markov chain to mix to the stationary distribution and the first $M < T$ samples are often dropped as burn-in samples. However, to keep the computational complexity the same as the previous computational methods, we set $T = L$ and $M = 0$. The first choice of transition distribution is $\sigma_t(\theta_t|\theta_{t-1}) = \mathcal{U}(\mathbb{S}^{d-1})$ which leads to independent Metropolis-Hasting (IMH). The second choice is $\sigma_t(\theta_t|\theta_{t-1}) = \mathrm{vMF}(\theta_t|\theta_{t-1}, \kappa)$ (the von Mises-Fisher distribution [17] with location $\theta_{t-1}$) which leads to random walk Metropolis-Hasting (RMH). Since both above transition distributions are symmetric $\sigma_t(\theta_t|\theta_{t-1}) = \sigma_t(\theta_{t-1}|\theta_t)$, the acceptance probability turns into:

$$\alpha = \min\left(1, \frac{f(\mathrm{W}_p^p(\theta_t'\sharp\mu, \theta_t'\sharp\nu)))}{f(\mathrm{W}_p^p(\theta_{t-1}\sharp\mu, \theta_{t-1}\sharp\nu)))}\right)$$

which means that the acceptance probability equals 1 and $\theta_t'$ is always accepted as $\theta_t$ if it can increase the energy function. We refer to the IMH estimation and the RMH estimation of the EBSW distance as IMH-EBSW and RMH-EBSW in turn.

**Computational algorithms and complexities.** We refer the reader to Algorithm 5, Algorithm 6, and Algorithm 7 in Appendix B.3 for the detailed algorithms of the SIR-EBSW, the IMH-EBSW, and the RMH-EBSW. Without counting the complexity of the sampling algorithm, the time complexity and the space complexity of both the SIR and the MCMC estimation of EBSW are $\mathcal{O}(Ln\log n + Lnd)$ and $\mathcal{O}(L(n + d))$ which are the same as the IS-EBSW and the SW distance. However, the practical computational time and memory of the SIR and the MCMC estimation depend on the efficiency of implementing the sampling algorithm e.g., resampling and acceptance-rejection.

**Unbiasedness.** The SIR and the MCMC sampling do not give an unbiased estimation for $\mathrm{EBSW}_p^p(\mu,\nu; f)$. However, they are also unbiased estimations of the power $p$ of a valid distance. We refer to Appendix B.3 for detailed definitions and properties of the distance. Therefore, it is safe to use the approximation from the SIR and the MCMC.

**Gradient Estimation:** In the IS estimation, the expectation is with respect to the proposal distribution $\sigma_0(\theta)$ that does not depend on two input measures $\mu_\phi$. In the SIR estimation and the MCMC estimation, the expectation is with respect to the slicing distribution $\sigma_{\mu_\phi,\nu}(\theta, f)$ that depends on $\mu_\phi$. Therefore, the log-derivative trick (Reinforce) should be used to derive the gradient estimation. We give the detailed derivation in Appendix B.3. However, the log-derivative trick is often unstable in practice. A simpler solution is to create the slicing distribution from an independent copy of $\mu_\phi$. In particular, we denote $\mu_{\phi'}$ is the independent copy of $\mu_\phi$ with $\phi'$ equals $\phi$ in terms of value. Therefore, we can obtain the slicing distribution $\sigma_{\mu_{\phi'},\nu}(\theta; f)$ that does not depend on $\mu_\phi$. This approach still gives the same value of distance, we refer to Appendix B.3 for a more careful discussion.

## 4 Experiments

In this section, we first visualize the shape of the energy-based slicing distribution in a simple case. After that, we focus on showing the favorable performance of the EBSW compared to the other sliced Wasserstein variants in point-cloud gradient flows, color transfer, and deep point-cloud reconstruction.

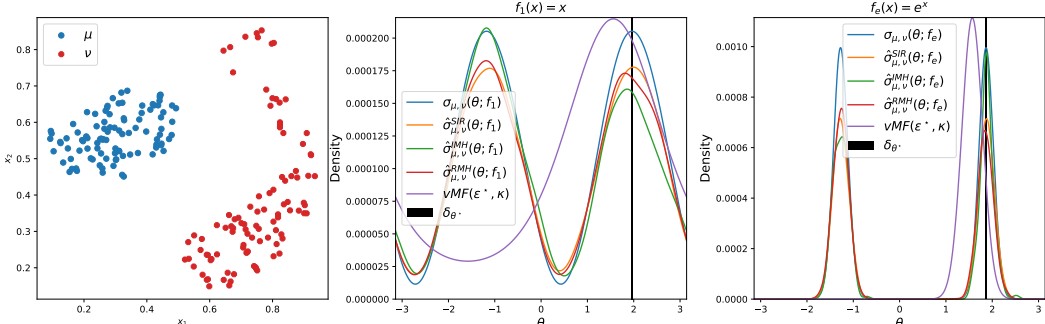

Figure 1: Visualization of the true and the sampled energy-based slicing distributions, the optimal vMF distribution from the v-DSW, and the max projecting direction from the Max-SW.

In experiments, we denote EBSW-e for the exponential energy function i.e., $f(x) = e^x$, and EBSW-1 for the identity energy function i.e., $f(x) = x$. We use $p = 2$ for all sliced Wasserstein variants.

## 4.1 Visualization of energy-based slicing distribution

We visualize the shape of the energy-based slicing distribution in two dimensions in Figure 1. In particular, we consider comparing two empirical distributions in the left-most figure (taken from [11]). We utilize the SIR, the IMH, and the RMH to obtain $10^4$ Monte Carlo samples from the energy-based slicing distribution. For the IMH and the RHM, we burn in the $10^4$ samples. After that, we use the von Mises kernel density estimation to obtain the density function. We also present the ground-truth density of the energy-based slicing distribution by uniform discretizing the unit-sphere. Moreover, we also show the optimal vMF distribution from the DSW (tuning $\kappa \in \{1, 5, 10, 50, 100\}$) and the "max" projecting direction from the Max-SW ($T = 100$, step size 0.1). The middle figure is according to the energy function $f_1(x) = x$ and the right-most figure is according to the energy function $f_e(x) = e^x$. From the figures, we observe that all sampling methods can approximate well the true slicing distribution. In contrast, the vMF distribution from v-DSW is misspecified to approximate the energy distribution, and the "max" projecting direction from the Max-SW can capture only one mode. We also observe that the exponential energy function makes the density more concentrated around the modes than the identity energy function (polynomial of degree 1).

## 4.2 Point-Cloud Gradient Flows

We model a distribution $\mu(t)$ flowing with time $t$ along the gradient flow of a loss functional $\mu(t) \to \mathcal{D}(\mu(t), \nu)$ [40] that drives it towards a target distribution $\nu$ where $\mathcal{D}$ is our sliced Wasserstein variants. We consider discrete flows, namely. we set $\nu = \frac{1}{n} \sum_{i=1}^{n} \delta_{Y_i}$ as a fixed empirical target distribution and the model distribution $\mu(t) = \frac{1}{n} \sum_{i=1}^{n} \delta_{X_i(t)}$. Here, the model distribution is parameterized by a time-varying point cloud $X(t) = (X_i(t))_{i=1}^{n} \in \left(\mathbb{R}^d\right)^n$. Starting from an initial condition at time $t = 0$, we integrate the ordinary differential equation $\dot{X}(t) = -n \nabla_{X(t)} \left[\mathcal{D}\left(\frac{1}{n} \sum_{i=1}^{n} \delta_{X_i(t)}, \nu\right)\right]$ for each iteration. We choose $\mu(0)$ and $\nu$ are two point-cloud shapes in ShapeNet Core-55 dataset [4]. After that, we solve the flows by using the Euler scheme with 500 iterations and step size 0.0001.

**Quantitative Results.** We show the Wasserstein-2 distance between $\mu(t)$ and $\nu$ ($t \in \{0, 100, 200, 300, 400, 500\}$) and the computational time of the SW variants in Table 1. Here, we set $L = 100$ for SW, and EBSW variants. For the Max-SW we set $T = 100$, and report the best result for the step size for finding the max projecting direction in $\{0.001, 0.01, 0.1\}$. For the v-DSW, we report the best result for $(L, T) \in \{(10, 10), (50, 2), (2, 50)\}$, $\kappa \in \{1, 10, 50\}$, and the learning rate for finding the location in $\{0.001, 0.01, 0.1\}$. We observe that the IS-EBSW-e helps to drive the flow to converge faster than baselines. More importantly, the computational time of the IS-EBSW-e is approximately the same as the SW and is faster than both the Max-SW and the v-DSW. We report more detailed experiments with other EBSW variants and different settings of hyperparameters $(L, T)$ in Table 3 in Appendix C.1. From the additional experiments, we see that the EBSW-e variants give lower Wasserstein-2 distances than the baseline with the same scaling of complexity (same $L/T$). Despite having comparable performance, the SIR-EBSW-e, the IMH-EBSW-e, and the RMH-EBSW-e

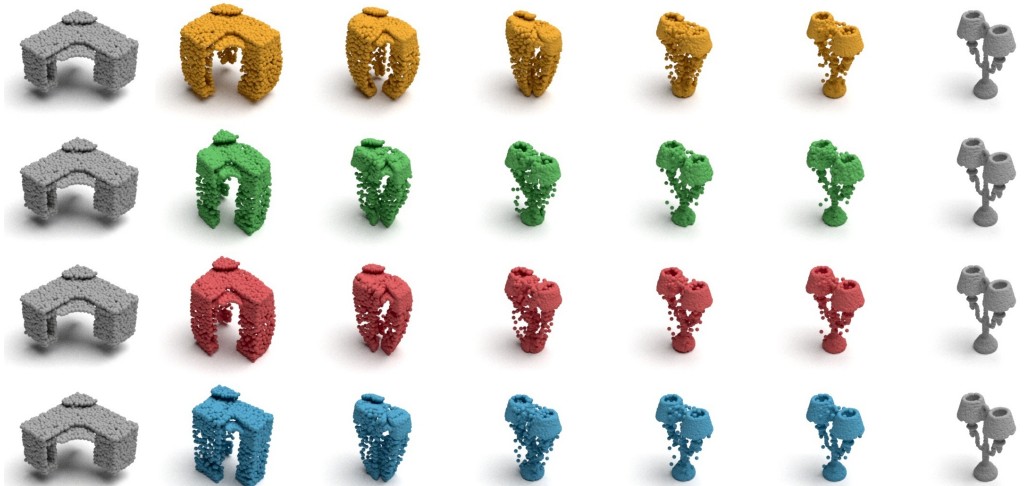

Figure 2: Gradient flows from the SW, the Max-SW, the v-DSW, and the IS-EBSW-e in turn.

Table 1: Summary of Wasserstein-2 scores [12] (multiplied by $10^4$) from three different runs, computational time in second (s) to reach step 500 of different sliced Wasserstein variants in gradient flows.

| Distances | Step 0 ($W_2 \downarrow$) | Step 100 ($W_2 \downarrow$) | Step 200 ($W_2 \downarrow$) | Step 300 ($W_2 \downarrow$) | Step 400 ($W_2 \downarrow$) | Step 500 ($W_2 \downarrow$) | Time (s $\downarrow$) |
|---|---|---|---|---|---|---|---|
| SW | $2048.29 \pm 0.0$ | $986.93 \pm 9.55$ | $350.66 \pm 5.32$ | $99.69 \pm 1.85$ | $27.03 \pm 0.65$ | $9.41 \pm 0.27$ | $\mathbf{17.06 \pm 0.45}$ |
| Max-SW | $2048.29 \pm 0.0$ | $506.56 \pm 9.28$ | $93.54 \pm 3.39$ | $22.2 \pm 0.79$ | $9.62 \pm 0.22$ | $6.83 \pm 0.22$ | $28.38 \pm 0.05$ |
| v-DSW | $2048.29 \pm 0.0$ | $649.33 \pm 8.77$ | $127.4 \pm 5.06$ | $29.44 \pm 1.25$ | $10.95 \pm 1.0$ | $5.68 \pm 0.56$ | $21.2 \pm 0.02$ |
| IS-EBSW-e | $2048.29 \pm 0.0$ | $\mathbf{419.09 \pm 2.64}$ | $\mathbf{71.02 \pm 0.46}$ | $\mathbf{18.2 \pm 0.05}$ | $\mathbf{6.9 \pm 0.08}$ | $\mathbf{3.3 \pm 0.08}$ | $17.63 \pm 0.02$ |

($\kappa = 10$) are slower than the IS-EBSW-e variant. Also, we see that the EBSW-e variants are better than the EBSW-1 variants. We refer to Table 4 for comparing gradient estimators of the EBSW.

**Qualitative Results.** We show the point-cloud flows of the SW, the Max-SW, the v-DSW, and the IS-EBSW-e, in Figure 2. The flows from the SIR-EBSW-e, the IMG-EBSW-e, and the RMH-EBSW-e are added in Figure 4 in the Appendix C.1. From the figure, the transitions of the flows from the EBSW-e variants are smoother to the target than other baselines.

### 4.3 Color Transfer

We build a gradient flow that starts from the empirical distribution over the normalized color palette (RGB) of the source image to the empirical distribution over the normalized color palette (RGB) of the target image. Since the value of the color palette is in the set $\{0, \ldots, 255\}^3$, we must do an additional rounding step at the final step of the Euler scheme with 2000 steps and step size 0.0001.

**Results.** We use the same setting for the SW variants as in the previous section. We show both transferred images, corresponding computational times, and Wasserstein-2 distances in Figure 3. We observe the same phenomenon as in the previous section, namely, the IS-EBSW-e variants perform the best in terms of changing the color of the source image to the target in the Wasserstein-2 metric while the computational time is only slightly higher. Moreover, the transferred image from the IS-EBSW-e is visually more similar to the target image in color than other baselines, namely, it has a less orange color. We refer to Figure 5 in Appendix C.2 for additional experiments including the results for the SIR-EBSW-e, the IMH-EBSW-e, the RMH-EBSW-e, the results for the identity energy function, the results for changing hyperparamters $(L, T)$, and the results for comparing gradient estimators. Overall, we observe similar phenomenons as in the previous gradient flow section.

### 4.4 Deep Point-Cloud Reconstruction

We follow [33] to train point-cloud autoencoders with sliced Wasserstein distances on the ShapeNet Core-55 dataset [4]. In short, we aim to estimate an autoencoder that contains an encoder $f_\phi$ that maps a point cloud $X \in \mathbb{R}^{nd}$ to a latent code $z \in \mathbb{R}^h$, and a decoder $g_\psi$ that maps the latent

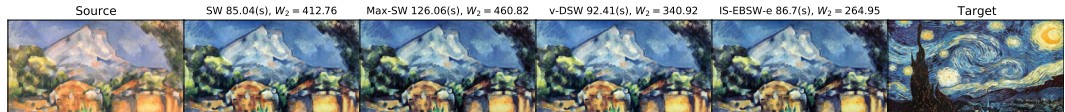

Figure 3: The figures show the source image, the target image, the transferred images from sliced Wasserstein variants, the corresponding Wasserstein-2 distances to the target color palette, and the computational time.

Table 2: Reconstruction errors from three different runs of autoencoders trained by different distances. The sliced Wasserstein distance and the Wasserstein distance are multiplied by 100.

| Distance | Epoch 20 | | Epoch 100 | | Epoch 200 | |
|---|---|---|---|---|---|---|
| | $SW_2(\downarrow)$ | $W_2(\downarrow)$ | $SW_2(\downarrow)$ | $W_2(\downarrow)$ | $SW_2(\downarrow)$ | $W_2(\downarrow)$ |
| SW | $2.97 \pm 0.14$ | $12.67 \pm 0.18$ | $2.29 \pm 0.04$ | $10.63 \pm 0.05$ | $2.15 \pm 0.04$ | $9.97 \pm 0.08$ |
| Max-SW | $2.91 \pm 0.06$ | $12.33 \pm 0.05$ | $2.24 \pm 0.05$ | $10.40 \pm 0.06$ | $2.14 \pm 0.10$ | $9.84 \pm 0.12$ |
| v-DSW | $2.84 \pm 0.02$ | $12.64 \pm 0.02$ | $2.21 \pm 0.01$ | $10.52 \pm 0.04$ | $2.07 \pm 0.09$ | $9.81 \pm 0.05$ |
| IS-EBSW-e | $\mathbf{2.68 \pm 0.03}$ | $\mathbf{11.90 \pm 0.04}$ | $\mathbf{2.18 \pm 0.04}$ | $\mathbf{10.27 \pm 0.01}$ | $\mathbf{2.04 \pm 0.09}$ | $\mathbf{9.69 \pm 0.14}$ |

code $z$ to a reconstructed point cloud $\tilde{X} \in \mathbb{R}^{nd}$. We want to have the pair $f_\phi$ and $g_\psi$ such that $\tilde{X} = g_\psi(f_\phi(X)) \approx X$ for all $X \sim p(X)$ which is our data distribution. To do that, we solve the following optimization problem: $\min_{\phi,\gamma} \mathbb{E}_{X \sim \mu(X)}[\mathcal{S}(P_X, P_{g_\gamma(f_\phi(X))})]$, where $\mathcal{S}$ is a sliced Wasserstein variant, and $P_X$ denotes the empirical distribution over the point cloud $X$. The backbone for the autoencoder is a variant of Point-Net [37] with an embedding size of 256. We train the autoencoder for 200 epochs using an SGD optimizer with a learning rate of 1e-3, a batch size of 128, a momentum of 0.9, and a weight decay of 5e-4. We give more detail in Appendix C.3

**Quantitative Results.** We evaluate the trained autoencoders on a different dataset: ModelNet40 dataset [46] using two distances: sliced Wasserstein distance ($L = 1000$), and the Wasserstein distance. We follow the same hyper-parameters settings as the previous sections and show the reconstruction errors at epochs 20, 100, and 200 from the SW, the Max-SW, the DSW, and the IS-EBSW-e in Table 2. The reconstruction errors are the average of corresponding distances on all point-clouds. From the table, we observe that the IS-EBSW-e can help to train the autoencoder faster in terms of the Wasserstein distance and the sliced Wasserstein distance. We refer to Table 5 in Appendix C.3 for similar ablation studies as in the previous sections including the results for the SIR-EBSW-e, the IMH-EBSW-e, the RMH-EBSW-e, the results for the identity energy function, the results for changing hyperparameters $(L, T)$, and the results for comparing gradient estimators.

**Qualitative Results.** We show some ground-truth point-clouds ModelNet40 and their corresponding reconstructed point-clouds from different models ($L = 100$) at epochs 200 and 20 in Figure 6- 7 respectively. Overall, the qualitative results are visually consistent with the quantitative results.

# 5   Limitations and Conclusion

**Limitations.** The first limitation of EBSW is that its MCMC variants are not directly parallelizable, resulting in slow computation. Additionally, a universal effective choice of the energy-based function is an open question. In the paper, we use simple and computationally effective energy-based functions, such as the exponential function and the polynomial function. Finally, proving the triangle inequality of EBSW is challenging due to the expressiveness of the energy-based slicing distribution.

**Conclusion.** We have presented a new variant of sliced Wasserstein distance, named energy-based sliced Wasserstein (EBSW) distance. The key ingredient of the EBSW is the energy-based slicing distribution which has a density at a projecting direction proportional to an increasing function of the one-dimensional Wasserstein value of that direction. We provide theoretical properties of the EBSW including the topological properties, and statistical properties. Moreover, we propose to compute the EBSW with three different techniques including importance sampling, sampling importance resampling, and Markov Chain Monte Carlo. Also, we discuss the computational properties of different techniques. Finally, we demonstrate the favorable performance of the EBSW compared to existing projecting directions selection sliced Wasserstein variants by conducting experiments on point-cloud gradient flows, color transfer, and deep point-cloud reconstruction.

## Acknowledgements

We would like to thank Peter Mueller for his insightful discussion during the course of this project. NH acknowledges support from the NSF IFML 2019844 and the NSF AI Institute for Foundations of Machine Learning.

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

# Supplement to "Energy-Based Sliced Wasserstein Distance"

We first provide skipped proofs in the main text in Appendix A. We then provide additional materials including additional background, detailed algorithms, and discussion in Appendix B. Additional experimental results in point-cloud gradient flows, color transfer, and deep point-cloud reconstruction in Appendix C. Finally, we report the computational infrastructure in Appendix D.

## A    Proofs

### A.1    Proof of Theorem 1

**Non-negativity and Symmetry.** the non-negativity and symmetry properties of the EBSW follow directly by the non-negativity and symmetry of the Wasserstein distance since it is an expectation of the one-dimensional Wasserstein distance.

**Identity.** We need to show that $\text{EBSW}_p(\mu, \nu; f) = 0$ if and only if $\mu = \nu$. First, from the definition of EBSW, we obtain directly $\mu = \nu$ implies $\text{EBSW}_p(\mu, \nu; f) = 0$. For the reverse direction, we use the same proof technique in [3]. If $\text{EBSW}_p(\mu, \nu; f) = 0$, we have $\int_{\mathbb{S}^{d-1}} W_p(\theta \sharp \mu, \theta \sharp \nu) \, d\sigma_{\mu,\nu}(\theta; f, p) = 0$. Hence, we have $W_p(\theta \sharp \mu, \theta \sharp \nu) = 0$ for $\sigma_{\mu,\nu}(\theta; f, p)$-almost surely $\theta \in \mathbb{S}^{d-1}$. Since $\sigma_{\mu,\nu}(\theta; f, p)$ is continuous, we have $W_p(\theta \sharp \mu, \theta \sharp \nu) = 0$ for all $\theta \in \mathbb{S}^{d-1}$. From the identity property of the Wasserstein distance, we obtain $\theta \sharp \mu = \theta \sharp \nu$ for $\sigma_{\mu,\nu}(\theta; f, p)$-a.e $\theta \in \mathbb{S}^{d-1}$. Therefore, for any $t \in \mathbb{R}$ and $\theta \in \mathbb{S}^{d-1}$, we have:

$$\mathcal{F}[\mu](t\theta) = \int_{\mathbb{R}^d} e^{-it\langle \theta, x \rangle} d\mu(x) = \int_{\mathbb{R}} e^{-itz} d\theta \sharp \mu(z) = \mathcal{F}[\theta \sharp \mu](t)$$

$$= \mathcal{F}[\theta \sharp \nu](t) = \int_{\mathbb{R}} e^{-itz} d\theta \sharp \nu(z) = \int_{\mathbb{R}^d} e^{-it\langle \theta, x \rangle} d\nu(x) = \mathcal{F}[\nu](t\theta),$$

where $\mathcal{F}[\gamma](w) = \int_{\mathbb{R}^{d'}} e^{-i\langle w, x \rangle} d\gamma(x)$ denotes the Fourier transform of $\gamma \in \mathcal{P}(\mathbb{R}^{d'})$. By the injectivity of the Fourier transform, we obtain $\mu = \nu$ which concludes the proof.

### A.2    Proof of Proposition 1

(a) We first provide the proof for the inequality $\text{SW}_p(\mu, \nu) \leq \text{EBSW}_p(\mu, \nu; f)$. It is equivalent to prove that

$$\mathbb{E}_{\theta \sim \mathcal{U}(\mathbb{S}^{d-1})} \left[ W_p^p(\theta \sharp \mu, \theta \sharp \nu) \right] \leq \mathbb{E}_{\theta \sim \sigma_{\mu,\nu}(\theta; f, p)} \left[ W_p^p(\theta \sharp \mu, \theta \sharp \nu) \right].$$

From the law of large number, it is sufficient to demonstrate that

$$\frac{1}{L} \sum_{i=1}^{L} W_p^p(\theta_i \sharp \mu, \theta_i \sharp \nu) \leq \sum_{i=1}^{L} \frac{W_p^p(\theta_i \sharp \mu, \theta_i \sharp \nu) f(W_p^p(\theta_i \sharp \mu, \theta_i \sharp \nu))}{\sum_{i=1}^{L} f(W_p^p(\theta_i \sharp \mu, \theta_i \sharp \nu))}, \tag{9}$$

for any $L \geq 1$ and $\theta_1, \ldots, \theta_L \overset{\text{i.i.d.}}{\sim} \mathcal{U}(\mathbb{S}^{d-1})$. To ease the presentation, we denote $a_i = W_p^p(\theta_i \sharp \mu, \theta_i \sharp \nu)$ and $b_i = f(W_p^p(\theta_i \sharp \mu, \theta_i \sharp \nu))$ for all $1 \leq i \leq L$. The inequality (9) becomes:

$$\frac{1}{L} \left( \sum_{i=1}^{L} a_i \right) \left( \sum_{i=1}^{L} b_i \right) \leq \sum_{i=1}^{L} a_i b_i. \tag{10}$$

We prove the inequality (10) via an induction argument. It is clear that this inequality holds when $L = 1$. We assume that this inequality holds for any $L$. We now verify that the inequality (10) also holds for $L + 1$. Without loss of generality, we assume that $a_1 \leq a_2 \leq \ldots \leq a_L \leq a_{L+1}$. Since the function $f$ is an increasing function, it indicates that $b_1 \leq b_2 \leq \ldots \leq b_L \leq b_{L+1}$. Applying the induction hypothesis for $a_1, \ldots, a_L$ and $b_1, \ldots, b_L$, we find that

$$\left( \sum_{i=1}^{L} a_i \right) \left( \sum_{i=1}^{L} b_i \right) \leq L \sum_{i=1}^{L} a_i b_i.$$

This inequality leads to

$$(\sum_{i=1}^{L+1} a_i)(\sum_{i=1}^{L+1} b_i) \leq L \sum_{i=1}^{L} a_i b_i + (\sum_{i=1}^{L} a_i) b_{L+1} + (\sum_{i=1}^{L} b_i) a_{L+1} + a_{L+1} b_{L+1}$$

Therefore, to obtain the conclusion of the hypothesis for $L + 1$, it is sufficient to demonstrate that

$$L \sum_{i=1}^{L} a_i b_i + (\sum_{i=1}^{L} a_i) b_{L+1} + (\sum_{i=1}^{L} b_i) a_{L+1} + a_{L+1} b_{L+1} \leq (L+1)(\sum_{i=1}^{L+1} a_i b_i),$$

which is equivalent to show that

$$(\sum_{i=1}^{L} a_i) b_{L+1} + (\sum_{i=1}^{L} b_i) a_{L+1} \leq \sum_{i=1}^{L} a_i b_i + L a_{L+1} b_{L+1}. \tag{11}$$

Since $a_{L+1} \geq a_i$ and $b_{L+1} \geq b_i$ for all $1 \leq i \leq L$, we have $(a_{L+1} - a_i)(b_{L+1} - b_i) \geq 0$, which is equivalent to $a_{L+1} b_{L+1} + a_i b_i \geq a_{L+1} b_i + b_{L+1} a_i$ for all $1 \leq i \leq L$. By taking the sum of these inequalities over $i$ from 1 to $L$, we obtain the conclusion of inequality (11). Therefore, we obtain the conclusion of the induction argument for $L + 1$, which indicates that inequality (10) holds for all $L$. As a consequence, we obtain the inequality $\text{SW}_p(\mu, \nu) \leq \text{EBSW}_p(\mu, \nu; f)$.

(b) We recall the definition of the Max-SW:

$$\text{Max-SW}_p(\mu, \nu) = \max_{\theta \in \mathbb{S}^{d-1}} W_p(\theta \sharp \mu, \theta \sharp \nu).$$

Since $\mathbb{S}^{d-1}$ is compact and the function $\theta \to W_p(\theta \sharp \mu, \theta \sharp \nu)$ is continuous, we have $\theta^\star = \text{argmax}_{\theta \in \mathbb{S}^{d-1}} W_p(\theta \sharp \mu, \theta \sharp \nu)$. From Definition 2, for any $p \geq 1$, dimension $d \geq 1$, energy-function $f$, and $\mu, \nu \in \mathcal{P}_p(\mathbb{R}^d)$ we have:

$$\text{EBSW}_p(\mu, \nu) = \left(\mathbb{E}_{\theta \sim \sigma_{\mu,\nu}(\theta; f, p)}\left[W_p^p(\theta \sharp \mu, \theta \sharp \nu)\right]\right)^{\frac{1}{p}}$$
$$\leq \left(\mathbb{E}_{\theta \sim \sigma_{\mu,\nu}(\theta; f, p)}\left[W_p^p(\theta^\star \sharp \mu, \theta^\star \sharp \nu)\right]\right)^{\frac{1}{p}} = W_p^p(\theta^* \sharp \mu, \theta^* \sharp \nu) = \text{Max-SW}_p(\mu, \nu).$$

Furthermore, by applying the Cauchy-Schwartz inequality, we have:

$$\text{Max-SW}_p^p(\mu, \nu) = \max_{\theta \in \mathbb{S}^{d-1}} \left(\inf_{\pi \in \Pi(\mu,\nu)} \int_{\mathbb{R}^d} |\theta^\top x - \theta^\top y|^p \, d\pi(x, y)\right)$$
$$\leq \max_{\theta \in \mathbb{S}^{d-1}} \left(\inf_{\pi \in \Pi(\mu,\nu)} \int_{\mathbb{R}^d \times \mathbb{R}^d} \|\theta\|^p \|x - y\|^p d\pi(x, y)\right)$$
$$= \inf_{\pi \in \Pi(\mu,\nu)} \int_{\mathbb{R}^d \times \mathbb{R}^d} \|\theta\|^p \|x - y\|^p d\pi(x, y)$$
$$= \inf_{\pi \in \Pi(\mu,\nu)} \int_{\mathbb{R}^d \times \mathbb{R}^d} \|x - y\|^p d\pi(x, y)$$
$$\leq \inf_{\pi \in \Pi(\mu,\nu)} \int_{\mathbb{R}^d \times \mathbb{R}^d} |x - y|^p d\pi(x, y)$$
$$= W_p^p(\mu, \nu),$$

after taking the $p$-rooth, we completes the proof.

## A.3  Proof of Theorem 2

We aim to show that for any sequence of probability measures $(\mu_k)_{k \in \mathbb{N}}$ and $\mu$ in $\mathcal{P}_p(\mathbb{R}^d)$, $\lim_{k \to +\infty} \text{EBSW}_p(\mu_k, \mu; \bar{f}) = 0$ if and only if for any continuous and bounded function $f : \mathbb{R}^d \to \mathbb{R}$, $\lim_{k \to +\infty} \int f \, d\mu_k = \int f \, d\mu$.

We now show that $\lim_{k \to \infty} \text{EBSW}_p(\mu_k, \mu; \bar{f}) = 0$ implies $(\mu_k)_{k \in \mathbb{N}}$ converges weakly to $\mu$. Let $(\mu_k)_{k \in \mathbb{N}}$ be a sequence such that $\lim_{k \to \infty} \text{EBSW}_p(\mu_k, \mu; \bar{f}) = 0$, we suppose $(\mu_k)_{k \in \mathbb{N}}$ does not converge weakly to $\mu$. So, let $\mathcal{D}_\mathcal{P}$ be the Lévy-Prokhorov metric, $\lim_{k \to \infty} \mathcal{D}_{\mathcal{P}(\mu_k, \mu)} \neq 0$ that implies

there exists $\varepsilon > 0$ and a subsequence $\left(\mu_{\psi(k)}\right)_{k\in\mathbb{N}}$ with an increasing function $\psi : \mathbb{N} \to \mathbb{N}$ such that for any $k \in \mathbb{N}$: $\mathcal{D}_{\mathcal{P}}(\mu_{\psi(k)}, \mu) \geq \varepsilon$. Using the Holder inequality with $\mu, \nu \in \mathbb{P}_p(\mathbb{R}^d)$, we have:

$$
\begin{aligned}
\text{EBSW}_p(\mu, \nu; \bar{f}) &= \left( \mathbb{E}_{\theta \sim \sigma_{\mu,\nu}(\theta; \bar{f}, p)} \left[ W_p^p \left( \theta\sharp\mu, \theta\sharp\nu \right) \right] \right)^{\frac{1}{p}} \\
&\geq \mathbb{E}_{\theta \sim \sigma_{\mu,\nu}(\theta; \bar{f}, p)} \left[ W_p \left( \theta\sharp\mu, \theta\sharp\nu \right) \right] \\
&\geq \mathbb{E}_{\theta \sim \mathcal{U}(\mathbb{S}^{d-1})} \left[ W_p \left( \theta\sharp\mu, \theta\sharp\nu \right) \right] \\
&\geq \mathbb{E}_{\theta \sim \mathcal{U}(\mathbb{S}^{d-1})} \left[ W_1 \left( \theta\sharp\mu, \theta\sharp\nu \right) \right] \\
&= \text{SW}_1(\mu, \nu).
\end{aligned}
$$

Therefore, $\lim_{k\to\infty} \text{SW}_1(\mu_{\psi(k)}, \mu) = 0$ which implies that there exists a subsequence $\left(\mu_{\phi(\psi(k))}\right)_{k\in\mathbb{N}}$ with an increasing function $\phi : \mathbb{N} \to \mathbb{N}$ such that $\left(\mu_{\phi(\psi(k))}\right)_{k\in\mathbb{N}}$ converges weakly to $\mu$ by Lemma S1 in [27]. Therefore a contradiction appears from the assumption of $\lim_{k\to\infty} \mathcal{D}_{\mathcal{P}}\left(\mu_{\phi(\psi(k))}, \mu\right) \neq 0$. Therefore, $\lim_{k\to\infty} \text{EBSW}_p(\mu_k, \mu; \bar{f}) = 0$, $(\mu_k)_{k\in\mathbb{N}}$ converges weakly to $\mu$.

When $(\mu_k)_{k\in\mathbb{N}}$ converges weakly to $\mu$, we have $(\theta\sharp\mu_k)_{k\in\mathbb{N}}$ converges weakly to $\theta\sharp\mu$ for any $\theta \in \mathbb{S}^{d-1}$ by the continuous mapping theorem. From [43], the weak convergence implies the convergence under the Wasserstein distance. So, we have $\lim_{k\to\infty} W_p(\theta\sharp\mu_k, \mu) = 0$. Moreover, using the fact that the Wasserstein distance is also bounded, hence, the bounded convergence theorem implies:

$$
\begin{aligned}
\lim_{k\to\infty} \text{EBSW}_p^p(\mu_k, \mu; \bar{f}) &= \lim_{k\to\infty} \mathbb{E}_{\theta \sim \sigma_{\mu,\nu}(\theta; \bar{f}, p)} \left[ W_p^p \left( \theta\sharp\mu_k, \theta\sharp\mu \right) \right] \\
&= \lim_{k\to\infty} \frac{\mathbb{E}_{\theta \sim \sigma_0(\theta)} \left[ \text{W}_p^p(\theta\sharp\mu_k, \theta\sharp\mu) w_{\mu_k, \mu, \sigma_0, \bar{f}, p}(\theta) \right]}{\mathbb{E}_{\theta \sim \sigma_0(\theta)} \left[ w_{\mu_k, \mu, \sigma_0, \bar{f}, p}(\theta) \right]} \\
&= 0.
\end{aligned}
$$

Again, using the continuous mapping theorem with function $x \to x^{1/p}$, we have $\lim_{k\to\infty} \text{EBSW}_p(\mu_k, \mu; \bar{f}) \to 0$. We conclude the proof.

### A.4  Proof of Proposition 2

We first show that the following inequality holds

$$
\mathbb{E}[\text{Max-SW}_p(\mu_n, \mu)] \leq C\sqrt{(d+1)\log n/n}
$$

where $C > 0$ is some universal constant and the outer expectation is taken with respect to the random variables $X_1, \ldots, X_n$. We now follow the proof technique from in [30]. Let $F_{n,\theta}^{-1}$ and $F_\theta^{-1}$ be the inverse cumulative distributions of two push-forward measures $\theta\sharp\mu_n$ and $\theta\sharp\mu$. Using the closed-form expression of the Wasserstein distance in one dimension, we obtain the following equations and inequalities:

$$
\begin{aligned}
\text{Max-SW}_p^p(\mu_n, \mu) &= \max_{\theta \in \mathbb{S}^{d-1}} \int_0^1 |F_{n,\theta}^{-1}(u) - F_\theta^{-1}(u)|^p du \\
&= \max_{\theta \in \mathbb{R}^d : \|\theta\|=1} \int_0^1 |F_{n,\theta}^{-1}(u) - F_\theta^{-1}(u)|^p du \\
&\leq \text{diam}(\mathcal{X}) \max_{\theta \in \mathbb{R}^d : \|\theta\| \leq 1} |F_{n,\theta}(x) - F_\theta(x)|^p.
\end{aligned}
$$

where $\mathcal{X} \subset \mathbb{R}^d$ is the compact set of the probability measure $\mu$. We can check that

$$
\max_{\theta \in \mathbb{R}^d : \|\theta\| \leq 1} |F_{n,\theta}(x) - F_\theta(x)| = \sup_{A \in \mathcal{H}} |\mu_n(A) - \mu(A)|,
$$

where $\mathcal{H}$ is the set of half-spaces $\{z \in \mathbb{R}^d : \theta^\top z \leq x\}$ for all $\theta \in \mathbb{R}^d$ such that $\|\theta\| \leq 1$. From VC inequality (Theorem 12.5 in [9]), we have:

$$
\mathbb{P}\left( \sup_{A \in \mathcal{H}} |\mu_n(A) - \mu(A)| > t \right) \leq 8S(\mathcal{H}, n)e^{-nt^2/32}.
$$

with $S(\mathcal{H}, n)$ is the growth function which is upper bounded by $(n+1)^{VC(\mathcal{H})}$ due to the Sauer Lemma (Proposition 4.18 in "High-Dimensional Statistics: A Non-Asymptotic Viewpoint", Wainwright et al). From Example 4.21 in [44], we get $VC(\mathcal{H}) = d + 1$.

Let $8S(\mathcal{H}, n)e^{-nt^2/32} \leq \delta$, we have $t^2 \geq \frac{32}{n} \log\left(\frac{8S(\mathcal{H},n)}{\delta}\right)$. Therefore, we obtain

$$\mathbb{P}\left(\sup_{A \in \mathcal{H}} |\mu_n(A) - \mu(A)| \leq \sqrt{\frac{32}{n} \log\left(\frac{8S(\mathcal{H},n)}{\delta}\right)}\right) \geq 1 - \delta,$$

Using the Jensen inequality and the tail sum expectation for non-negative random variable, we have:

$$\mathbb{E}\left[\sup_{A \in \mathcal{H}} |\mu_n(A) - \mu(A)|\right]$$

$$\leq \sqrt{\mathbb{E}\left[\sup_{A \in \mathcal{H}} |\mu_n(A) - \mu(A)|\right]^2} = \sqrt{\int_0^\infty \mathbb{P}\left(\left(\sup_{A \in \mathcal{H}} |\mu_n(A) - \mu(A)|\right)^2 > t\right) dt}$$

$$= \sqrt{\int_0^u \mathbb{P}\left(\left(\sup_{A \in \mathcal{H}} |\mu_n(A) - \mu(A)|\right)^2 > t\right) dt + \int_u^\infty \mathbb{P}\left(\left(\sup_{A \in \mathcal{H}} |\mu_n(A) - \mu(A)|\right)^2 > t\right) dt}$$

$$\leq \sqrt{\int_0^u 1 dt + \int_u^\infty 8S(\mathcal{H}, n)e^{-nt/32} dt} = \sqrt{u + 256S(\mathcal{H}, n)\frac{e^{-nu/32}}{n}}.$$

Since the inequality holds for any $u$, we optimize for $u$. Let $f(u) = u + 256S(\mathcal{H}, n)\frac{e^{-nu/32}}{n}$, we have $f'(u) = 1 + 8S(\mathcal{H}, n)e^{-nu/32}$. Setting $f'(u) = 0$, we obtain the minima $u^\star = \frac{32 \log(8S(\mathcal{H},n))}{n}$. Therefore, we have:

$$\mathbb{E}\left[\sup_{A \in \mathcal{H}} |\mu_n(A) - \mu(A)|\right] \leq \sqrt{f(u^\star)} \leq \sqrt{\frac{32 \log(8S(\mathcal{H}, n))}{n} + 32} \leq C\sqrt{\frac{(d+1)\log(n+1)}{n}},$$

by using Sauer Lemma. As a consequence, we obtain:

$$\mathbb{E}[\text{EBSW}_p(\mu_n, \mu; f)] \leq C\sqrt{(d+1)\log n/n},$$

which completes the proof.

## B    Additional Materials

### B.1    Additional Background

**Sliced Wasserstein.** When two probability measures are empirical probability measures on $n$ supports: $\mu = \frac{1}{n}\sum_{i=1}^n \delta_{x_i}$ and $\nu = \frac{1}{n}\sum_{i=1}^n \delta_{y_i}$, the SW distance can be computed by sorting projected supports. In particular, we have $\theta\sharp\mu = \frac{1}{n}\sum_{i=1}^n \delta_{\theta^\top x_i}$, $\theta\sharp\nu = \frac{1}{n}\sum_{i=1}^n \delta_{\theta^\top y_i}$, and $\text{W}_p^p(\theta\sharp\mu, \theta\sharp\nu) = \frac{1}{n}\sum_{i=1}^n (\theta^\top x_{(i)} - \theta^\top y_{(i)})^p$ where $\theta^\top x_{(i)}$ is the ordered projected supports. We provide the pseudo-code for computing the SW in Algorithm 1.

**Max sliced Wasserstein.** The Max-SW is often computed by the projected gradient ascent. The sub-gradient is used when the one-dimensional optimal matching is not unique e.g., in discrete cases. We provide the projected (sub)-gradient ascent algorithm for computing the Max-SW in Algorithm 2. Compared to the SW, the Max-SW needs two hyperparameters which are the number of iterations $T$ and the step size $\eta$. Moreover, the empirical estimation of the Max-SW might not converge to the Max-SW when $T \to \infty$.

**Distributional sliced Wasserstein.** To solve the optimization of the DSW, we need to use the stochastic (sub)-gradient ascent algorithm. In particular, we first need to estimate the gradient:

$$\nabla_\psi \left(\mathbb{E}_{\theta \sim \sigma_\psi(\theta)} \text{W}_p^p(\theta\sharp\mu, \theta\sharp\nu)\right)^{\frac{1}{p}} = \frac{1}{p}\left(\mathbb{E}_{\theta \sim \sigma_\psi(\theta)} \text{W}_p^p(\theta\sharp\mu, \theta\sharp\nu)\right)^{\frac{1-p}{p}} \nabla_\psi \mathbb{E}_{\theta \sim \sigma_\psi(\theta)} \text{W}_p^p(\theta\sharp\mu, \theta\sharp\nu).$$

---
**Algorithm 1** Computational algorithm of the SW distance
---
**Input:** Probability measures $\mu$ and $\nu$, $p \geq 1$, and the number of projections $L$.
**for** $l = 1$ to $L$ **do**
    Sample $\theta_l \sim \mathcal{U}(\mathbb{S}^{d-1})$
    Compute $v_l = \mathrm{W}_p(\theta_l \sharp \mu, \theta_l \sharp \nu)$
**end for**
Compute $\widehat{SW}_p(\mu, \nu; L) = \left( \frac{1}{L} \sum_{l=1}^{L} v_l \right)^{\frac{1}{p}}$
**Return:** $\widehat{SW}_p(\mu, \nu; L)$

---

---
**Algorithm 2** Computational algorithm of the Max-SW distance
---
**Input:** Probability measures $\mu$ and $\nu$, $p \geq 1$, the number of iterations $T$, and the step size $\eta$.
Sample $\hat{\theta}_0 \sim \mathcal{U}(\mathbb{S}^{d-1})$
**for** $t = 1$ to $T$ **do**
    Compute $\hat{\theta}_t = \hat{\theta}_{t-1} + \eta \nabla_{\hat{\theta}_{t-1}} \mathrm{W}_p(\hat{\theta}_{t-1} \sharp \mu, \hat{\theta}_{t-1} \sharp \nu)$
    Compute $\hat{\theta}_t = \frac{\hat{\theta}_t}{||\hat{\theta}_t||_2}$
**end for**
Compute $\widehat{\text{Max-SW}}_p(\mu, \nu; T) = \mathrm{W}_p(\hat{\theta}_T \sharp \mu, \hat{\theta}_T \sharp \nu)$
**Return:** $\widehat{\text{Max-SW}}_p(\mu, \nu; T)$

---

---
**Algorithm 3** Computational algorithm of the DSW distance
---
**Input:** Probability measures $\mu$ and $\nu$, $p \geq 1$, the number of projections $L$, the number of iterations $T$, and the step size $\eta$.
Initialize $\hat{\psi}_0$
**for** $t = 1$ to $T$ **do**
    $\nabla_\psi = 0$
    **for** $l = 1$ to $L$ **do**
        Sample $\theta_{l,\psi} \sim \sigma_{\hat{\psi}_{t-1}(\theta)}$ via reparameterization.
        Compute $\hat{\theta}_t = \frac{\hat{\theta}_t}{||\hat{\theta}_t||_2}$
    **end for**
    Compute $\hat{\psi}_t = \hat{\psi}_{t-1} + \eta \frac{1}{p} \left( \frac{1}{L} \sum_{l=1}^{L} \mathrm{W}_p^p(\theta_{l,\psi} \sharp \mu, \theta_{l,\psi} \sharp \nu) \right)^{\frac{1-p}{p}} \frac{1}{L} \sum_{l=1}^{l} \nabla_\psi \mathrm{W}_p^p(\theta_{l,\psi} \sharp \mu, \theta_{l,\psi} \sharp \nu))$
**end for**
**for** $l = 1$ to $L$ **do**
    Sample $\theta_l \sim \sigma_{\hat{\psi}_T(\theta)}$ via reparameterization.
**end for**
Compute $\widehat{\text{DSW}}_p(\mu, \nu; T, L) = \left( \frac{1}{L} \sum_{l=1}^{L} \mathrm{W}_p^p(\theta_l \sharp \mu, \theta_l \sharp \nu) \right)^{\frac{1}{p}}$
**Return:** $\widehat{\text{DSW}}_p(\mu, \nu; T, L)$

---

To estimate the gradient $\nabla_\psi \mathbb{E}_{\theta \sim \sigma_\psi(\theta)} \mathrm{W}_p^p(\theta \sharp \mu, \theta \sharp \nu)$, we need to use reparameterization trick for $\sigma_\psi(\theta)$ e.g., the vMF distribution. After using the reparameterization trick, we can approximate the gradient $\nabla_\psi \mathbb{E}_{\theta \sim \sigma_\psi(\theta)} \mathrm{W}_p^p(\theta \sharp \mu, \theta \sharp \nu) = \frac{1}{L} \sum_{l=1}^{l} \nabla_\psi \mathrm{W}_p^p(\theta_{l,\psi} \sharp \mu, \theta_{l,\psi} \sharp \nu)$ where $\theta_{1,\psi}, \ldots, \theta_{L,\psi}$ are i.i.d reparameterized samples from $\sigma_\psi(\theta)$. Similarly, we approximiate $\mathbb{E}_{\theta \sim \sigma_\psi(\theta)} \mathrm{W}_p^p(\theta \sharp \mu, \theta \sharp \nu) = \frac{1}{L} \sum_{l=1}^{L} \mathrm{W}_p^p(\theta_l \sharp \mu, \theta_l \sharp \nu)$ . We refer to the details in the following papers [6, 32]. We review the algorithm for computing the DSW in Algorithm 3. Compared to the SW, the DSW needs three hyperparameters i.e., the number of projections $L$, the number of iterations $T$, and the step size $\eta$.

**Minimum Distance Estimator and Gradient Estimation.** In statistical inference, we are given the empirical samples $X_1, \ldots, X_n$ from the interested distribution $\nu$. Since we do not know the form of $\nu$, we might want to find an alternative representation. In particular, we want to find the best member $\mu_\phi$ in a family of distribution parameterized by $\phi \in \Phi$. To do that, we want to minimize the

**Algorithm 4** Computational algorithm of the IS-EBSW distance

---

**Input:** Probability measures $\mu$ and $\nu$, $p \geq 1$, the number of projections $L$, the energy function $f$.
**for** $l = 1$ to $L$ **do**
    Sample $\theta_l \sim \mathcal{U}(\mathbb{S}^{d-1})$
    Compute $v_l = \mathrm{W}_p(\theta_l \sharp \mu, \theta_l \sharp \nu)$
    Compute $w_l = f(\mathrm{W}_p(\theta_l \sharp \mu, \theta_l \sharp \nu))$
**end for**
Compute $\widehat{\mathrm{IS\text{-}EBSW}}_p(\mu, \nu; L, f) = \left( \sum_{l=1}^L v_l \frac{w_l}{\sum_{i=1}^L w_i} \right)^{\frac{1}{p}}$
**Return:** $\widehat{\mathrm{IS\text{-}EBSW}}_p(\mu, \nu; L, f)$

---

distance between $\mu_\phi$ and the empirical distribution $\nu_n = \frac{1}{n} \sum_{i=1}^n \delta_{X_i}$. This framework is named the minimum distance estimator [45]:

$\min_{\phi \in \Phi} \mathcal{D}(\mu_\phi, \nu_n)$, where $\mathcal{D}$ is a discrepancy between two distributions. The gradient ascent algorithm is often used to solve the problem. To do so, we need to compute the gradient $\nabla_\phi \mathcal{D}(\mu_\phi, \nu_n)$. When using sliced Wasserstein distances, the gradient $\nabla_\phi \mathcal{D}(\mu_\phi, \nu_n)$ is often approximated by a stochastic gradient since the SW distances involve an intractable expectation. In previous SW variants, the expectation does not depend on $\phi$, hence, we can use directly the Leibniz rule to exchange the gradient and the expectation, then perform the Monte Carlo approximation. In particular, we have $\nabla_\phi \mathbb{E}_{\theta \sim \sigma(\theta)}[\mathrm{W}_p^p(\theta \sharp \mu, \theta \sharp \nu)] = \mathbb{E}_{\theta \sim \sigma(\theta)}[\nabla_\phi \mathrm{W}_p^p(\theta \sharp \mu, \theta \sharp \nu)] \approx \frac{1}{L} \sum_{l=1}^L \nabla_\phi \mathrm{W}_p^p(\theta_l \sharp \mu, \theta_l \sharp \nu)$ for $\theta_1, \ldots, \theta_L \overset{i.i.d}{\sim} \sigma(\theta)$.

## B.2 Importance Sampling

**Derivation.** We first provide the derivation of the importance sampling estimation of EBSW. From the definition of the EBSW, we have:

$$
\begin{aligned}
\mathrm{EBSW}_p(\mu, \nu; f) &= \left( \mathbb{E}_{\theta \sim \sigma_{\mu,\nu}(\theta; f, p)} \left[ \mathrm{W}_p^p(\theta \sharp \mu, \theta \sharp \nu) \right] \right)^{\frac{1}{p}} \\
&= \left( \frac{\int_{\mathbb{S}^{d-1}} \mathrm{W}_p^p(\theta \sharp \mu, \theta \sharp \nu) f(\mathrm{W}_p^p(\theta \sharp \mu, \theta \sharp \nu)) d\theta}{\int_{\mathbb{S}^{d-1}} f(\mathrm{W}_p^p(\theta \sharp \mu, \theta \sharp \nu)) d\theta} \right)^{\frac{1}{p}} \\
&= \left( \frac{\int_{\mathbb{S}^{d-1}} \mathrm{W}_p^p(\theta \sharp \mu, \theta \sharp \nu) \frac{f(\mathrm{W}_p^p(\theta \sharp \mu, \theta \sharp \nu))}{\sigma_0(\theta)} \sigma_0(\theta) d\theta}{\int_{\mathbb{S}^{d-1}} \frac{f(\mathrm{W}_p^p(\theta \sharp \mu, \theta \sharp \nu))}{\sigma_0(\theta)} \sigma_0(\theta) d\theta} \right)^{\frac{1}{p}} \\
&= \left( \frac{\mathbb{E}_{\theta \sim \sigma_0(\theta)} \left[ \mathrm{W}_p^p(\theta \sharp \mu, \theta \sharp \nu) \frac{f(\mathrm{W}_p^p(\theta \sharp \mu, \theta \sharp \nu))}{\sigma_0(\theta)} \right]}{\mathbb{E}_{\theta \sim \sigma_0(\theta)} \left[ \frac{f(\mathrm{W}_p^p(\theta \sharp \mu, \theta \sharp \nu))}{\sigma_0(\theta)} \right]} \right)^{\frac{1}{p}} \\
&= \left( \frac{\mathbb{E}_{\theta \sim \sigma_0(\theta)} \left[ \mathrm{W}_p^p(\theta \sharp \mu, \theta \sharp \nu) w_{\mu, \nu, \sigma_0}(\theta) \right]}{\mathbb{E}_{\theta \sim \sigma_0(\theta)} \left[ w_{\mu, \nu, \sigma_0}(\theta) \right]} \right)^{\frac{1}{p}}.
\end{aligned}
$$

**Algorithms.** We provide the algorithm for the IS estimation of the EBSW in Algorithm 4. Compared to the algorithm of the SW in Algorithm 1, the IS-EBSW can be obtained by only adding one or two lines of code in practice. Therefore, the computation of the IS-EBSW is as fast as the SW while being more meaningful.

**Gradient Estimators.** Let $\mu_\phi$ be parameterized by $\phi$, we derive now the gradient estimator $\nabla_\phi \mathrm{EBSW}_p(\mu, \nu; f)$ through importance sampling. We have:

$$
\begin{aligned}
\nabla_\phi \mathrm{EBSW}_p(\mu_\phi, \nu; f) = \frac{1}{p} & \left( \frac{\mathbb{E}_{\theta \sim \sigma_0(\theta)} \left[ \mathrm{W}_p^p(\theta \sharp \mu_\phi, \theta \sharp \nu) w_{\mu_\phi, \nu, \sigma_0, f}(\theta) \right]}{\mathbb{E}_{\theta \sim \sigma_0(\theta)} \left[ w_{\mu_\phi, \nu, \sigma_0, f}(\theta) \right]} \right)^{\frac{1-p}{p}} \\
& \nabla_\phi \frac{\mathbb{E}_{\theta \sim \sigma_0(\theta)} \left[ \mathrm{W}_p^p(\theta \sharp \mu_\phi, \theta \sharp \nu) w_{\mu_\phi, \nu, \sigma_0, f}(\theta) \right]}{\mathbb{E}_{\theta \sim \sigma_0(\theta)} \left[ w_{\mu_\phi, \nu, \sigma_0, f}(\theta) \right]}.
\end{aligned}
$$

We denote $A(\phi) = \mathbb{E}_{\theta \sim \sigma_0(\theta)} \left[ W_p^p(\theta \sharp \mu_\phi, \theta \sharp \nu) w_{\mu_\phi, \nu, \sigma_0, f}(\theta) \right]$, $B(\phi) = \mathbb{E}_{\theta \sim \sigma_0(\theta)} \left[ w_{\mu_\phi, \nu, \sigma_0, f}(\theta) \right]$, we have

$$\nabla_\phi \frac{A(\phi)}{B(\phi)} = \frac{B(\phi) \nabla_\phi A(\phi) - A(\phi) \nabla \phi B(\phi)}{B^2(\phi)}.$$

Using Monte Carlo samples $\theta_1, \ldots, \theta_L \sim \sigma_0(\theta)$ after using the Lebnitz rule to exchange the differentiation and the expectation, we obtain:

$$\left( \frac{\mathbb{E}_{\theta \sim \sigma_0(\theta)} \left[ W_p^p(\theta \sharp \mu_\phi, \theta \sharp \nu) w_{\mu_\phi, \nu, \sigma_0, f}(\theta) \right]}{\mathbb{E}_{\theta \sim \sigma_0(\theta)} \left[ w_{\mu_\phi, \nu, \sigma_0, f}(\theta) \right]} \right)^{\frac{1-p}{p}} \approx \left( \frac{\frac{1}{L} \sum_{l=1}^{L} \left[ W_p^p(\theta_l \sharp \mu_\phi, \theta_l \sharp \nu) w_{\mu_\phi, \nu, \sigma_0, f}(\theta_l) \right]}{\frac{1}{L} \sum_{l=1}^{L} \left[ w_{\mu_\phi, \nu, \sigma_0, f}(\theta_l) \right]} \right)^{\frac{1-p}{p}},$$

$$\nabla_\phi A(\phi) \approx \frac{1}{L} \sum_{l=1}^{L} \nabla_\phi \left( W_p^p(\theta_l \sharp \mu_\phi, \theta_l \sharp \nu) w_{\mu_\phi, \nu, \sigma_0, f}(\theta) \right),$$

$$\nabla_\phi B(\phi) \approx \frac{1}{L} \sum_{l=1}^{L} \nabla_\phi w_{\mu_\phi, \nu, \sigma_0, f}(\theta),$$

which yields the gradient estimation. If we construct the slicing distribution by using a copy of $\mu_\phi$ i.e., $\mu_{\phi'}$ with $\phi' = \phi$ in terms of value, the gradient estimator can be derived by:

$$\nabla_\phi \text{EBSW}_p(\mu_\phi, \nu; f) = \frac{1}{p} \left( \frac{\mathbb{E}_{\theta \sim \sigma_0(\theta)} \left[ W_p^p(\theta \sharp \mu_\phi, \theta \sharp \nu) w_{\mu_{\phi'}, \nu, \sigma_0, f}(\theta) \right]}{\mathbb{E}_{\theta \sim \sigma_0(\theta)} \left[ w_{\mu_{\phi'}, \nu, \sigma_0, f}(\theta) \right]} \right)^{\frac{1-p}{p}}$$
$$\frac{\nabla_\phi \mathbb{E}_{\theta \sim \sigma_0(\theta)} \left[ W_p^p(\theta \sharp \mu_\phi, \theta \sharp \nu) w_{\mu_{\phi'}, \nu, \sigma_0, f}(\theta) \right]}{\mathbb{E}_{\theta \sim \sigma_0(\theta)} \left[ w_{\mu_{\phi'}, \nu, \sigma_0, f}(\theta) \right]},$$

Using Monte Carlo samples $\theta_1, \ldots, \theta_L \sim \sigma_0(\theta)$ after using the Lebnitz rule to exchange the differentiation and the expectation, we obtain:

$$\left( \frac{\mathbb{E}_{\theta \sim \sigma_0(\theta)} \left[ W_p^p(\theta \sharp \mu_\phi, \theta \sharp \nu) w_{\mu_{\phi'}, \nu, \sigma_0, f}(\theta) \right]}{\mathbb{E}_{\theta \sim \sigma_0(\theta)} \left[ w_{\mu_{\phi'}, \nu, \sigma_0, f}(\theta) \right]} \right)^{\frac{1-p}{p}} \approx \left( \frac{\frac{1}{L} \sum_{l=1}^{L} \left[ W_p^p(\theta_l \sharp \mu_\phi, \theta_l \sharp \nu) w_{\mu_{\phi'}, \nu, \sigma_0, f}(\theta_l) \right]}{\frac{1}{L} \sum_{l=1}^{L} \left[ w_{\mu_{\phi'}, \nu, \sigma_0, f}(\theta_l) \right]} \right)^{\frac{1-p}{p}},$$

$$\nabla_\phi \mathbb{E}_{\theta \sim \sigma_0(\theta)} \left[ W_p^p(\theta \sharp \mu_\phi, \theta \sharp \nu) w_{\mu_{\phi'}, \nu, \sigma_0, f}(\theta) \right] \approx \frac{1}{L} \sum_{l=1}^{L} \left( \nabla_\phi W_p^p(\theta_l \sharp \mu_\phi, \theta_l \sharp \nu) \right) w_{\mu_\phi, \nu', \sigma_0, f}(\theta),$$

$$\mathbb{E}_{\theta \sim \sigma_0(\theta)} \left[ w_{\mu_{\phi'}, \nu, \sigma_0, f}(\theta) \right] \approx \frac{1}{L} \sum_{l=1}^{L} w_{\mu_{\phi'}, \nu, \sigma_0, f}(\theta).$$

It is worth noting that using a copy of $\mu_\phi$ does not change the value of the distance. This trick will show its true benefit when dealing with the SIR, and the MCMC methods. However, we still discuss it in the IS case for completeness. We refer to the "copy" trick is the "parameter-copy" gradient estimator while the original one is the conventional estimator.

**Importance Weighted sliced Wasserstein distance.** Although the IS estimation of the EBSW is not an unbiased estimation for finite $L$, it is an unbiased estimation of a valid distance on the space of probability measures. We refer to the distance as the importance weighted sliced Wasserstein distance (IWSW) which has the following definition.

**Definition 3.** *For any $p \geq 1$, dimension $d \geq 1$, energy function $f$, a continuous proposal distribution $\sigma_0(\theta) \sim \mathcal{P}(\mathbb{S}^{d-1})$ and two probability measures $\mu \in \mathcal{P}_p(\mathbb{R}^d)$ and $\nu \in \mathbb{R}^d$, the importance weighted sliced Wasserstein (IWSW) distance is defined as follows:*

$$IWSW_p(\mu, \nu; f) = \left( \mathbb{E} \left[ \frac{\frac{1}{L} \sum_{l=1}^{L} \left[ W_p^p(\theta_l \sharp \mu, \theta_l \sharp \nu) w_{\mu, \nu, \sigma_0, f, p}(\theta_l) \right]}{\frac{1}{L} \sum_{l=1}^{L} \left[ w_{\mu, \nu, \sigma_0, f, p}(\theta_l) \right]} \right] \right)^{\frac{1}{p}}, \tag{12}$$

*where the expectation is with respect to $\theta_1, \ldots, \theta_L \overset{i.i.d}{\sim} \sigma_0(\theta)$, and $w_{\mu, \nu, \sigma_0, f, p}(\theta) = \frac{f(W_p^p(\theta \sharp \mu, \theta \sharp \nu))}{\sigma_0(\theta)}$.*

**Algorithm 5** Computational algorithm of the SIR-EBSW distance

**Input:** Probability measures $\mu$ and $\nu$, $p \geq 1$, the number of projections $L$, the energy function $f$.
**for** $l = 1$ to $L$ **do**
    Sample $\theta_l \sim \mathcal{U}(\mathbb{S}^{d-1})$
    Compute $w_l = f(\mathrm{W}_p(\theta_l \sharp \mu, \theta_l \sharp \nu))$
**end for**
**for** $l = 1$ to $L$ **do**
    Compute $\hat{w}_l = \frac{f(\mathrm{W}_p(\theta_l \sharp \mu, \theta_l \sharp \nu))}{\sum_{i=1}^{L} f(\mathrm{W}_p(\theta_i \sharp \mu, \theta_i \sharp \nu))}$
**end for**
**for** $l = 1$ to $L$ **do**
    Sample $\theta_l \sim \mathrm{Cat}(\hat{w}_1, \ldots, \hat{w}_L)$
    Compute $v_l = \mathrm{W}_p(\theta_l \sharp \mu, \theta_l \sharp \nu)$
**end for**
Compute $\widehat{\mathrm{SIR\text{-}SW}}_p(\mu, \nu; L, f) = \left( \frac{1}{L} \sum_{l=1}^{L} v_l \right)^{\frac{1}{p}}$
**Return:** $\widehat{\mathrm{SIR\text{-}SW}}_p(\mu, \nu; L, f)$

---

**Algorithm 6** Computational algorithm of the SW distance and the IMH-EBSW distance

**Input:** Probability measures $\mu$ and $\nu$, $p \geq 1$, the number of projections $L$, the energy function $f$.
Sample $\theta_1 \sim \mathcal{U}(\mathbb{S}^{d-1})$
Compute $v_1 = \mathrm{W}_p(\theta_1 \sharp \mu, \theta_1 \sharp \nu)$
**for** $l = 2$ to $L$ **do**
    Sample $\theta_l' \sim \mathcal{U}(\mathbb{S}^{d-1})$
    Compute $\alpha = \min \left( 1, \frac{f(\mathrm{W}_p^p(\theta_l' \sharp \mu, \theta_l' \sharp \nu))}{f(\mathrm{W}_p^p(\theta_{l-1} \sharp \mu, \theta_{l-1} \sharp \nu))} \right)$
    Sample $u \sim \mathcal{U}([0, 1])$
    **if** $\alpha \geq u$ **then**
        Set $\theta_l = \theta_l'$
    **else if** $\alpha < u$ **then**
        Set $\theta_l = \theta_{l-1}$
    **end if**
    $v_l = \mathrm{W}_p(\theta_l \sharp \mu, \theta_l \sharp \nu)$
**end for**
Compute $\widehat{\mathrm{IMH\text{-}EBSW}}_p(\mu, \nu; L, f) = \left( \frac{1}{L} \sum_{l=1}^{L} v_l \right)^{\frac{1}{p}}$
**Return:** $\widehat{\mathrm{IMH\text{-}EBSW}}_p(\mu, \nu; L)$

---

The IWSW is semi-metric, it also does not suffer from the curse of dimensionality, and it induces weak convergence. The proofs can be derived by following directly the proofs of the EBSW in Appendix A.1, Apendix A.3, and Appendix A.4. Therefore, using the IS estimation of the EBSW is as safe as the SW.

## B.3  Sampling Importance Resampling and Markov Chain Monte Carlo

**Algorithms.** We first provide the algorithm for computing the EBSW via the SIR, the IMH, and the RMH in Algorithm 5-7.

**Gradient estimators.** We derive the reinforce gradient estimator of the EBSW for the SIR, the IMH, and the RHM sampling.

$$\nabla_\phi \mathrm{EBSW}_p(\mu_\phi, \nu; f) = \frac{1}{p} \left( \mathbb{E}_{\theta \sim \sigma_{\mu_\phi, \nu}(\theta; f)} \left[ \mathrm{W}_p^p(\theta \sharp \mu_\phi, \theta \sharp \nu) \right] \right)^{\frac{1-p}{p}} \nabla_\phi \mathbb{E}_{\theta \sim \sigma_{\mu_\phi, \nu}(\theta; f)} \left[ \mathrm{W}_p^p(\theta \sharp \mu_\phi, \theta \sharp \nu) \right].$$

We have:

$$\nabla_\phi \mathbb{E}_{\theta \sim \sigma_{\mu_\phi, \nu}(\theta; f)} \left[ \mathrm{W}_p^p(\theta \sharp \mu_\phi, \theta \sharp \nu) \right] = \mathbb{E}_{\theta \sim \sigma_{\mu_\phi, \nu; f}(\theta)} \left[ \mathrm{W}_p^p(\theta_\phi \sharp \mu, \theta \sharp \nu) \nabla_\phi \log \left( \mathrm{W}_p^p(\theta \sharp \mu_\phi, \theta \sharp \nu) \sigma_{\mu_\phi, \nu}(\theta; f) \right) \right]$$

**Algorithm 7** Computational algorithm of the SW distance and the RMH-EBSW distance

---

**Input:** Probability measures $\mu$ and $\nu$, $p \geq 1$, the number of projections $L$, the energy function $f$, the concentration parameter $\kappa$.

Sample $\theta_1 \sim \mathcal{U}(\mathbb{S}^{d-1})$

Compute $v_1 = \mathrm{W}_p(\theta_1 \sharp \mu, \theta_1 \sharp \nu)$

**for** $l = 2$ to $L$ **do**

   Sample $\theta_l' \sim \mathrm{vMF}(\theta_{l-1}, \kappa)$

   Compute $\alpha = \min\left(1, \frac{f(\mathrm{W}_p^p(\theta_l' \sharp \mu, \theta_l' \sharp \nu)))}{f(\mathrm{W}_p^p(\theta_{l-1} \sharp \mu, \theta_{l-1} \sharp \nu)))}\right)$

   Sample $u \sim \mathcal{U}([0,1])$

   **if** $\alpha \geq u$ **then**

     Set $\theta_l = \theta_l'$

   **else if** $\alpha < u$ **then**

     Set $\theta_l = \theta_{l-1}$

   **end if**

   $v_l = \mathrm{W}_p(\theta_l \sharp \mu, \theta_l \sharp \nu)$

**end for**

Compute $\widehat{\mathrm{RMH\text{-}EBSW}}_p(\mu, \nu; L, f) = \left(\frac{1}{L}\sum_{l=1}^{L} v_l\right)^{\frac{1}{p}}$

**Return:** $\widehat{\mathrm{RMH\text{-}EBSW}}_p(\mu, \nu; L)$

---

and

$$
\begin{aligned}
\nabla_\phi \log\left(\mathrm{W}_p^p(\theta \sharp \mu_\phi, \theta \sharp \nu)\sigma_{\mu_\phi, \nu}(\theta; f)\right) &= \nabla_\phi \log(\mathrm{W}_p^p \theta \sharp \mu_\phi, \theta \sharp \nu)) + \nabla_\phi \log(f(\mathrm{W}_p^p(\theta \sharp \mu_\phi, \theta \sharp \nu))) \\
&\quad - \nabla_\phi \log\left(\int_{\mathbb{S}^{d-1}} f(\mathrm{W}_p^p(\theta \sharp \mu_\phi, \theta \sharp \nu))d\theta\right) \\
&= \frac{1}{\mathrm{W}_p^p(\theta \sharp \mu_\phi, \theta \sharp \nu))}\nabla_\phi \mathrm{W}_p^p(\theta \sharp \mu_\phi, \theta \sharp \nu) \\
&\quad + \frac{1}{f(\mathrm{W}_p^p(\theta \sharp \mu_\phi, \theta \sharp \nu))}\nabla_\phi f(\mathrm{W}_p^p(\theta \sharp \mu_\phi, \theta \sharp \nu)) \\
&\quad - \nabla_\phi \log\left(\int_{\mathbb{S}^{d-1}} f(\mathrm{W}_p^p(\theta \sharp \mu_\phi, \theta \sharp \nu))d\theta\right),
\end{aligned}
$$

and

$$
\begin{aligned}
&\nabla_\phi \log\left(\int_{\mathbb{S}^{d-1}} f(\mathrm{W}_p^p(\theta \sharp \mu_\phi, \theta \sharp \nu))d\theta\right) \\
&= \nabla_\phi \log\left(\mathbb{E}_{\theta \sim \mathcal{U}(\mathbb{S}^{d-1})}\left[f(\mathrm{W}_p^p(\theta \sharp \mu_\phi, \theta \sharp \nu))\frac{2\pi^{d/2}}{\Gamma(d/2)}\right]\right) \\
&= \nabla_\phi \log\left(\mathbb{E}_{\theta \sim \mathcal{U}(\mathbb{S}^{d-1})}\left[f(\mathrm{W}_p^p(\theta \sharp \mu_\phi, \theta \sharp \nu))\right]\right) \\
&= \frac{1}{\mathbb{E}_{\theta \sim \mathcal{U}(\mathbb{S}^{d-1})}\left[f(\mathrm{W}_p^p(\theta \sharp \mu_\phi, \theta \sharp \nu))\right]}\nabla_\phi \mathbb{E}_{\theta \sim \mathcal{U}(\mathbb{S}^{d-1})}\left[f(\mathrm{W}_p^p(\theta \sharp \mu_\phi, \theta \sharp \nu)\right].
\end{aligned}
$$

Using $L$ Monte Carlo samples from the SIR (or the IMH or the RMH) to approximate the expectation $\mathbb{E}_{\theta \sim \sigma_{\mu_\phi, \nu}(\theta; f)}$, and $L$ samples from $\mathcal{U}(\mathbb{S}^{d-1})$ to approximate the expectation $\mathbb{E}_{\theta \sim \mathcal{U}(\mathbb{S}^{d-1})}$, we obtain the gradient estimator of the EBSW. However, the reinforce gradient estimator is unstable in practice, especially with the energy function $f_e(x) = e^x$. Therefore, we propose a more simple gradient estimator which is:

$$
\nabla_\phi \mathrm{EBSW}_p(\mu_\phi, \nu; f) \approx \frac{1}{p}\left(\mathbb{E}_{\theta \sim \sigma_{\mu_{\phi'}, \nu}(\theta; f)}\left[\mathrm{W}_p^p(\theta \sharp \mu_\phi, \theta \sharp \nu)\right]\right)^{\frac{1-p}{p}}\mathbb{E}_{\theta \sim \sigma_{\mu_{\phi'}, \nu}(\theta; f)}\left[\nabla_\phi \mathrm{W}_p^p(\theta \sharp \mu_\phi, \theta \sharp \nu)\right].
$$

The key is to use a copy of the parameter $\phi'$ for constructing the slicing distribution $\sigma_{\mu_{\phi'}, \nu}(\theta; f)$, hence, we can exchange directly the differentiation and the expectation. It is worth noting that using the copy also affects the gradient estimation, it does not change the value of the distance. We refer to the "copy" trick is the "parameter-copy" gradient estimator while the original one is the conventional estimator.

**Population distance.** The approximated values of $p$-power EBSW from using the SIR, the IMH, and the RMH can be all written as $\frac{1}{L}\sum_{l=1}^{L} W_p^p(\theta_l\sharp\mu, \theta_l\sharp\nu)$. Here, the distributions of $\theta_1,\ldots,\theta_L$ are different. Therefore, they are not an unbiased estimation of the $\mathrm{EBSW}_p^p(\mu,\nu;f)$. However, the population distance of the estimation can be defined as in Definition 4.

**Definition 4.** *For any $p \geq 1$, dimension $d \geq 1$, energy function $f$, and two probability measures $\mu \in \mathcal{P}_p(\mathbb{R}^d)$ and $\nu \in \mathbb{R}^d$, the projected sliced Wasserstein (PSW) distance is defined as follows:*

$$PSW_p(\mu,\nu;f) = \left(\mathbb{E}\left[\frac{1}{L}\sum_{l=1}^{L} W_p^p(\theta_l\sharp\mu, \theta\sharp\nu)\right]\right)^{\frac{1}{p}}, \tag{13}$$

*where the expectation is with respect to $(\theta_1,\ldots,\theta_L) \sim \sigma(\theta_1,\ldots,\theta_L)$ which is a distribution defined by the SIR (the IMH or the RHM).*

The PSW is a valid metric since it satisfies the triangle inequality in addition to the symmetry, the non-negativity, and the identity. In particular, given three probability measures $\mu_1, \mu_2, \mu_3 \in \mathcal{P}_p(\mathbb{R}^d)$ we have:

$$
\begin{aligned}
\mathrm{PSW}_p(\mu_1,\mu_3) &= \left(\mathbb{E}_{(\theta_{1:L})\sim\sigma(\theta_{1:L})}\left[\frac{1}{L}\sum_{l=1}^{L} W_p^p\left(\theta_l\sharp\mu_1, \theta_l\sharp\mu_3\right)\right]\right)^{\frac{1}{p}} \\
&\leq \left(\mathbb{E}_{(\theta_{1:L})\sim\sigma(\theta_{1:L})}\left[\frac{1}{L}\sum_{t=1}^{L}\left(W_p\left(\theta_l\sharp\mu_1, \theta_l\sharp\mu_2\right) + W_p\left(\theta_l\sharp\mu_2, \theta_l\sharp\mu_3\right)\right)^p\right]\right)^{\frac{1}{p}} \\
&\leq \left(\mathbb{E}_{(\theta_{1:L})\sim\sigma(\theta_{1:L})}\left[\frac{1}{L}\sum_{t=1}^{L} W_p^p\left(\theta_l\sharp\mu_1, \theta_l\sharp\mu_2\right)\right]\right)^{\frac{1}{p}} \\
&\quad + \left(\mathbb{E}_{(\theta_{1:L})\sim\sigma(\theta_{1:L})}\left[\frac{1}{L}\sum_{l=1}^{T} W_p^p\left(\theta_l\sharp\mu_2, \theta_l\sharp\mu_3\right)\right]\right)^{\frac{1}{p}} \\
&= \mathrm{PSW}_p(\mu_1,\mu_2) + \mathrm{PSW}_p(\mu_2,\mu_3),
\end{aligned}
$$

where the first inequality is due to the triangle inequality of Wasserstein distance and the second inequality is due to the Minkowski inequality. The PSW also does not suffer from the curse of dimensionality, and it induces weak convergence. The proofs can be derived by following directly the proofs of the EBSW in Appendix A.1, Apendix A.3, and Appendix A.4. Therefore, using the SIR, the IMH, and the RMH estimation of the EBSWs are as safe as the SW.

## C  Additional Experiments

In this section, we provide additional results for point-cloud gradient flows in Appendix C.1, color transfer in Appendix C.2, and deep point-cloud reconstruction in Appendix C.3.

### C.1  Point-Cloud Gradient Flows

We provide the full experimental results including the IS-EBSW, the SIR-EBSW, the IMH-EBSW, and the RMH-EBSW with both the exponential energy function and the identity energy function in Table 3. In the table, we also include the results for the number of projections $L = 10$. In Table 3, we use the conventional gradient estimator for the IS-EBSW while the "parameter-copy" estimator is used for other variants of the EBSW. Therefore, we also provide the ablation studies comparing the gradient estimators in Table 4 by adding the results for the "parameter-copy" estimator for the IS-EBSW and the conventional estimator for other variants. Experimental settings are the same as in the main text.

**Quantitative Results.** From the two tables, we observe that the IS-EBSW is the best variant of the EBSW in both performance and computational time. Also, we observe that the exponential energy function is better than the identity energy function in this application. It is worth noting that the EBSW variants of all computational methods and energy functions are better than the baselines in

Table 3: Summary of Wasserstein-2 scores (multiplied by $10^4$) from three different runs, computational time in second (s) to reach step 500 of different sliced Wasserstein variants in gradient flows.

| Distances | Step 0 ($W_2 \downarrow$) | Step 100 ($W_2 \downarrow$) | Step 200 ($W_2 \downarrow$) | Step 300 ($W_2 \downarrow$) | Step 400($W_2 \downarrow$) | Step 500 ($W_2 \downarrow$) | Time (s $\downarrow$) |
|---|---|---|---|---|---|---|---|
| SW L=100 | $2048.29 \pm 0.0$ | $986.93 \pm 9.55$ | $350.66 \pm 5.32$ | $99.69 \pm 1.85$ | $27.03 \pm 0.65$ | $9.41 \pm 0.27$ | $\mathbf{17.06 \pm 0.45}$ |
| Max-SW T=100 | $2048.29 \pm 0.0$ | $506.56 \pm 9.28$ | $93.54 \pm 3.39$ | $22.2 \pm 0.79$ | $9.62 \pm 0.22$ | $6.83 \pm 0.22$ | $28.38 \pm 0.05$ |
| v-DSW L*T=100 | $2048.29 \pm 0.0$ | $649.33 \pm 8.77$ | $127.4 \pm 5.06$ | $29.44 \pm 1.25$ | $10.95 \pm 1.0$ | $5.68 \pm 0.56$ | $21.2 \pm 0.02$ |
| DSW L*T=100 | $2048.29 \pm 0.0$ | $519.63 \pm 2.43$ | $101.99 \pm 1.44$ | $24.11 \pm 0.49$ | $8.85 \pm 0.16$ | $4.62 \pm 0.15$ | $24.32 \pm 0.03$ |
| IS-EBSW-e L=100 | $2048.29 \pm 0.0$ | $\mathbf{419.09 \pm 2.64}$ | $\mathbf{71.02 \pm 0.46}$ | $\mathbf{18.2 \pm 0.05}$ | $\mathbf{6.9 \pm 0.08}$ | $\mathbf{3.3 \pm 0.08}$ | $17.63 \pm 0.02$ |
| SIR-EBSW-e L=100 | $2048.29 \pm 0.0$ | $435.02 \pm 1.1$ | $85.26 \pm 0.11$ | $21.96 \pm 0.12$ | $7.9 \pm 0.22$ | $3.79 \pm 0.17$ | $29.8 \pm 0.04$ |
| IMH-EBSW-e L=100 | $2048.29 \pm 0.0$ | $460.19 \pm 3.46$ | $91.28 \pm 1.19$ | $23.35 \pm 0.52$ | $8.26 \pm 0.26$ | $3.93 \pm 0.14$ | $49.3 \pm 0.54$ |
| RMH-EBSW-e L=100 | $2048.29 \pm 0.0$ | $454.92 \pm 3.25$ | $87.92 \pm 0.69$ | $22.66 \pm 0.46$ | $8.14 \pm 0.31$ | $3.82 \pm 0.24$ | $62.5 \pm 0.09$ |
| IS-EBSW-1 L=100 | $2048.29 \pm 0.0$ | $692.63 \pm 7.21$ | $167.75 \pm 3.12$ | $41.8 \pm 0.93$ | $12.31 \pm 0.27$ | $5.35 \pm 0.1$ | $17.91 \pm 0.28$ |
| SIR-EBSW-1 L=100 | $2048.29 \pm 0.0$ | $704.08 \pm 2.75$ | $169.88 \pm 0.47$ | $41.85 \pm 0.28$ | $12.58 \pm 0.24$ | $5.64 \pm 0.18$ | $30.56 \pm 0.05$ |
| IMH-EBSW-1 L=100 | $2048.29 \pm 0.0$ | $715.97 \pm 4.49$ | $171.42 \pm 1.25$ | $42.05 \pm 0.42$ | $12.6 \pm 0.1$ | $5.63 \pm 0.06$ | $50.01 \pm 0.01$ |
| RMH-EBSW-1 L=100 | $2048.29 \pm 0.0$ | $712.11 \pm 1.64$ | $173.47 \pm 1.49$ | $42.94 \pm 0.4$ | $12.68 \pm 0.15$ | $5.54 \pm 0.09$ | $64.01 \pm 0.08$ |
| SW L=10 | $2048.29 \pm 0.0$ | $988.57 \pm 14.01$ | $351.63 \pm 2.63$ | $101.54 \pm 2.45$ | $28.19 \pm 1.04$ | $10.11 \pm 0.34$ | $\mathbf{3.84 \pm 0.04}$ |
| Max-SW T=10 | $2048.29 \pm 0.0$ | $525.72 \pm 7.35$ | $134.8 \pm 4.6$ | $34.07 \pm 0.34$ | $10.77 \pm 0.15$ | $7.36 \pm 0.31$ | $6.55 \pm 0.06$ |
| IS-EBSW-e L=10 | $2048.29 \pm 0.0$ | $519.73 \pm 8.63$ | $\mathbf{92.14 \pm 1.29}$ | $\mathbf{23.94 \pm 0.07}$ | $\mathbf{9.03 \pm 0.33}$ | $\mathbf{4.59 \pm 0.22}$ | $5.57 \pm 0.03$ |
| SIR-EBSW-e L=10 | $2048.29 \pm 0.0$ | $\mathbf{508.86 \pm 8.49}$ | $104.47 \pm 1.93$ | $28.27 \pm 0.68$ | $10.56 \pm 0.08$ | $5.61 \pm 0.16$ | $6.84 \pm 0.06$ |
| IMH-EBSW-e L=10 | $2048.29 \pm 0.0$ | $621.51 \pm 22.49$ | $131.75 \pm 7.09$ | $34.42 \pm 1.89$ | $11.55 \pm 0.38$ | $5.56 \pm 0.09$ | $8.41 \pm 0.04$ |
| RMH-EBSW-e L=10 | $2048.29 \pm 0.0$ | $642.87 \pm 5.25$ | $135.91 \pm 8.39$ | $36.11 \pm 2.13$ | $12.57 \pm 0.75$ | $5.94 \pm 0.31$ | $9.69 \pm 0.04$ |
| IS-EBSW-1 L=10 | $2048.29 \pm 0.0$ | $713.65 \pm 5.68$ | $177.16 \pm 1.19$ | $45.07 \pm 0.17$ | $13.6 \pm 0.26$ | $6.16 \pm 0.22$ | $5.69 \pm 0.0$ |
| SIR-EBSW-1 L=10 | $2048.29 \pm 0.0$ | $731.4 \pm 9.37$ | $181.28 \pm 5.05$ | $44.99 \pm 1.07$ | $13.59 \pm 0.51$ | $6.68 \pm 0.27$ | $6.9 \pm 0.03$ |
| IMH-EBSW-1 L=10 | $2048.29 \pm 0.0$ | $772.86 \pm 28.09$ | $199.29 \pm 7.02$ | $48.73 \pm 1.69$ | $14.1 \pm 0.49$ | $6.25 \pm 0.35$ | $8.61 \pm 0.02$ |
| RMH-EBSW-1 L=10 | $2048.29 \pm 0.0$ | $810.1 \pm 10.2$ | $212.11 \pm 9.53$ | $54.62 \pm 2.63$ | $15.44 \pm 0.93$ | $6.74 \pm 0.32$ | $9.86 \pm 0.06$ |

Table 4: Summary of Wasserstein-2 scores (multiplied by $10^4$) from three different runs, computational time in second (s) to reach step 500 of different sliced Wasserstein variants in gradient flows.

| Distances | Step 0 ($W_2 \downarrow$) | Step 100 ($W_2 \downarrow$) | Step 200 ($W_2 \downarrow$) | Step 300 ($W_2 \downarrow$) | Step 400($W_2 \downarrow$) | Step 500 ($W_2 \downarrow$) | Time (s $\downarrow$) |
|---|---|---|---|---|---|---|---|
| IS-EBSW-e L=100 (c) | $2048.29 \pm 0.0$ | $435.39 \pm 1.82$ | $85.31 \pm 0.44$ | $21.9 \pm 0.09$ | $7.81 \pm 0.06$ | $3.68 \pm 0.07$ | $17.51 \pm 0.01$ |
| IS-EBSW-1 L=100 (c) | $2048.29 \pm 0.0$ | $711.33 \pm 7.2$ | $170.69 \pm 2.91$ | $42.2 \pm 0.79$ | $12.62 \pm 0.2$ | $5.7 \pm 0.11$ | $17.72 \pm 0.02$ |
| SIR-EBSW-1 L=100 | $2048.29 \pm 0.0$ | $685.87 \pm 8.35$ | $166.39 \pm 2.65$ | $41.52 \pm 0.56$ | $12.29 \pm 0.32$ | $5.56 \pm 0.1$ | $44.51 \pm 0.16$ |
| IMH-EBSW-1 L=100 | $2048.29 \pm 0.0$ | $700.47 \pm 9.13$ | $173.25 \pm 1.26$ | $44.08 \pm 0.52$ | $13.03 \pm 0.18$ | $5.93 \pm 0.2$ | $63.83 \pm 0.02$ |
| RMH-EBSW-1 L=100 | $2048.29 \pm 0.0$ | $711.0 \pm 10.98$ | $175.76 \pm 1.45$ | $44.5 \pm 0.56$ | $13.39 \pm 0.13$ | $6.06 \pm 0.05$ | $77.32 \pm 0.2$ |
| IS-EBSW-e L=10 (c) | $2048.29 \pm 0.0$ | $524.69 \pm 7.38$ | $107.37 \pm 2.18$ | $28.46 \pm 0.35$ | $10.13 \pm 0.38$ | $4.93 \pm 0.37$ | $5.54 \pm 0.04$ |
| IS-EBSW-1 L=10 (c) | $2048.29 \pm 0.0$ | $729.53 \pm 6.74$ | $179.35 \pm 1.7$ | $45.03 \pm 0.79$ | $13.32 \pm 0.82$ | $6.15 \pm 0.46$ | $5.7 \pm 0.03$ |
| SIR-EBSW-1 L=10 | $2048.29 \pm 0.0$ | $762.23 \pm 9.66$ | $202.2 \pm 5.23$ | $56.48 \pm 1.55$ | $19.05 \pm 0.83$ | $10.42 \pm 0.53$ | $8.45 \pm 0.02$ |
| IMH-EBSW-1 L=10 | $2048.29 \pm 0.0$ | $762.67 \pm 14.63$ | $200.3 \pm 6.48$ | $54.28 \pm 1.17$ | $18.11 \pm 0.36$ | $9.29 \pm 0.26$ | $10.02 \pm 0.02$ |
| RMH-EBSW-1 L=10 | $2048.29 \pm 0.0$ | $817.92 \pm 23.86$ | $220.66 \pm 2.55$ | $60.15 \pm 1.53$ | $20.0 \pm 0.7$ | $9.8 \pm 0.36$ | $11.35 \pm 0.03$ |

terms of Wasserstein-2 distances at the last epoch. For all sliced Wasserstein variants, we see that reducing the number of projections leads to worsening performance which is consistent with previous studies in previous works [28, 19]. In Table 3, the IS-EBSW uses the conventional gradient estimator while the SIR-EBSW, the IMH-EBSW, and the RMH-EBSW use the "parameter-copy" estimator. Therefore, we report the IS-EBSW with the "parameter-copy" estimator and the SIR-EBSW, the IMH-EBSW, and the RMH-EBSW with the Reinforce estimator (conventional estimator) in Table 4. From the table, we observe the "parameter-copy" estimator is worse than the conventional estimator in the case of IS-EBSW. For the SIR-EBSW, the IMH-EBSW, and the RMH-EBSW, we cannot use the exponential energy function due to the numerically unstable Reinforce estimator. In the case of the identity energy function, the exponential energy function is also worse than the "parameter-copy" estimator. Therefore, we recommend to use the IS-EBSW-e with the conventional gradient estimator.

**Qualitative Results.** We provide the visualization of the gradient flows from SW (L=100), Max-SW (T=100), v-DSW (L=10,T=10), and all the EBSW-e variants in Figure 4. Overall, we see that EBSW-e variants give smoother flows than other baselines. Despite having slightly different quantitative scores due to the approximation methods, the visualization from the EBSW-e variants is consistent. Therefore, the energy-based slicing function helps to improve the convergence of the source point-cloud to the target point-cloud.

## C.2 Color Transfer

Similar to the point-cloud gradient flow, we follow the same experimental settings of color transfer in the main text. We provide the full experimental results including the IS-EBSW, the SIR-EBSW, the IMH-EBSW, and the RMH-EBSW with both the exponential energy function and the identity energy function, with both $L = 10$ and $L = 100$, and with both gradient estimators in Figure 5.

**Results.** From the figure, we observe that IMH-EBSW-e gives the best Wasserstein-2 distance among all EBSW variants. Between the exponential energy function and the identity energy function,

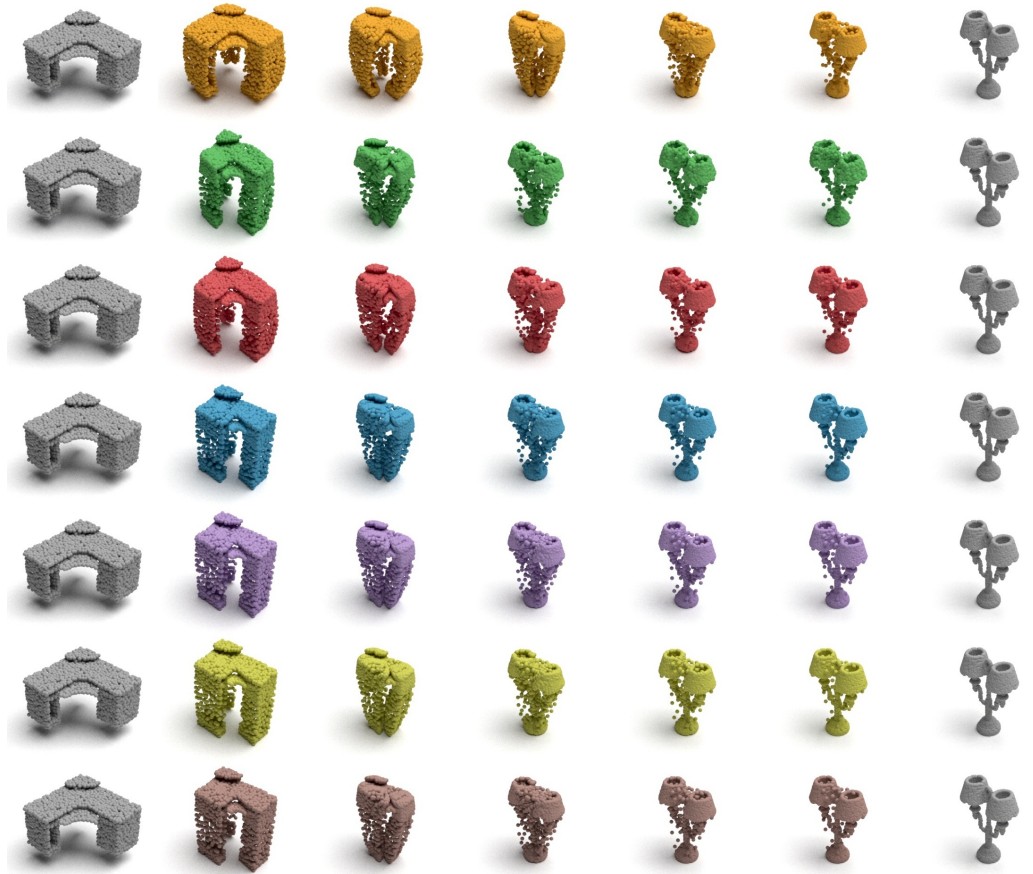

Figure 4: Gradient flows from the SW, the Max-SW, the v-DSW, the IS-EBSW-e, the SIR-EBSW-e, the IMH-EBSW-e, and the RMH-EBSW-e in turn.

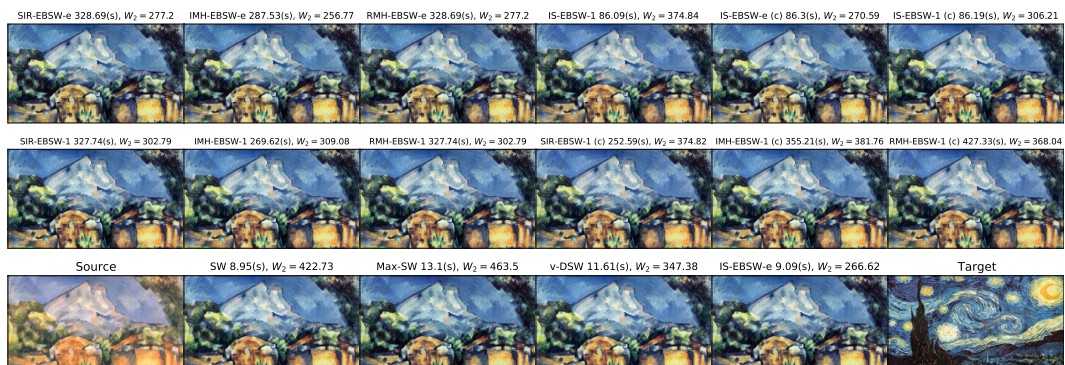

Figure 5: The first two rows are with $L = 100$, (c) denotes the "parameter-copy" (the SIR-EBSW-e, the IMH-EBSW-e, the RMH-EBSW always use the "parameter-copy" estimator since the conventional estimator is not stable for them), and the last row is with $L = 10$.

we see that the exponential energy function yields a better result for all EBSW variants. Similar to the gradient flow, reducing the number of projections to 10 also leads to worse results for all sliced Wasserstein variants For the gradient estimators, the conventional estimator is preferred for the IS-EBSW while the "parameter-copy" estimator is preferred for other EBSW variants.

Table 5: Reconstruction errors of different autoencoders measured by the (sliced) Wasserstein distance ($\times 100$). The results are from three different runs.

| Distance | Epoch 20 | | Epoch 100 | | Epoch 200 | |
|---|---|---|---|---|---|---|
| | $SW_2(\downarrow)$ | $W_2(\downarrow)$ | $SW_2(\downarrow)$ | $W_2(\downarrow)$ | $SW_2(\downarrow)$ | $W_2(\downarrow)$ |
| SW L=100 | $2.97 \pm 0.14$ | $12.67 \pm 0.18$ | $2.29 \pm 0.04$ | $10.63 \pm 0.05$ | $2.15 \pm 0.04$ | $9.97 \pm 0.08$ |
| Max-SW T=100 | $2.91 \pm 0.06$ | $12.33 \pm 0.05$ | $2.24 \pm 0.05$ | $10.40 \pm 0.06$ | $2.14 \pm 0.10$ | $9.84 \pm 0.12$ |
| v-DSW L*T=100 | $2.84 \pm 0.02$ | $12.64 \pm 0.02$ | $2.21 \pm 0.01$ | $10.52 \pm 0.04$ | $2.07 \pm 0.09$ | $9.81 \pm 0.05$ |
| DSW | $2.80$ | $12.59$ | $\mathbf{2.16}$ | $10.38$ | $2.07$ | $9.78$ |
| IS-EBSW-e L=100 | $\mathbf{2.68 \pm 0.03}$ | $\mathbf{11.90 \pm 0.04}$ | $2.18 \pm 0.04$ | $\mathbf{10.27 \pm 0.01}$ | $2.04 \pm 0.09$ | $\mathbf{9.69 \pm 0.14}$ |
| SIR-EBSW-e L=100 | $2.77 \pm 0.01$ | $12.16 \pm 0.04$ | $2.24 \pm 0.04$ | $10.40 \pm 0.01$ | $2.00 \pm 0.03$ | $9.72 \pm 0.04$ |
| IMH-EBSW-e L=100 | $2.75 \pm 0.03$ | $12.15 \pm 0.04$ | $2.19 \pm 0.08$ | $10.39 \pm 0.09$ | $\mathbf{1.99 \pm 0.05}$ | $9.72 \pm 0.10$ |
| RMH-EBSW-e L=100 | $2.83 \pm 0.02$ | $12.21 \pm 0.03$ | $2.20 \pm 0.03$ | $10.38 \pm 0.07$ | $2.02 \pm 0.02$ | $9.72 \pm 0.03$ |
| IS-EBSW-1 L=100 | $2.83 \pm 0.01$ | $12.37 \pm 0.01$ | $2.27 \pm 0.06$ | $10.59 \pm 0.07$ | $2.11 \pm 0.04$ | $9.90 \pm 0.02$ |
| SIR-EBSW-1 L=100 | $2.81 \pm 0.02$ | $12.32 \pm 0.03$ | $2.26 \pm 0.08$ | $10.56 \pm 0.14$ | $2.07 \pm 0.01$ | $9.81 \pm 0.08$ |
| IMH-EBSW-1 L=100 | $2.82 \pm 0.01$ | $12.32 \pm 0.02$ | $2.28 \pm 0.11$ | $10.55 \pm 0.13$ | $2.03 \pm 0.02$ | $9.81 \pm 0.02$ |
| RMH-EBSW-1 L=100 | $2.88 \pm 0.04$ | $12.42 \pm 0.06$ | $2.22 \pm 0.07$ | $10.37 \pm 0.06$ | $2.01 \pm 0.02$ | $9.73 \pm 0.02$ |
| SW L=10 | $2.99 \pm 0.12$ | $12.70 \pm 0.16$ | $2.30 \pm 0.01$ | $10.64 \pm 0.04$ | $2.17 \pm 0.06$ | $10.01 \pm 0.09$ |
| Max-SW T=10 | $3.00 \pm 0.07$ | $12.68 \pm 0.05$ | $2.31 \pm 0.08$ | $10.67 \pm 0.06$ | $2.14 \pm 0.04$ | $9.95 \pm 0.05$ |
| IS-EBSW-e L=10 | $\mathbf{2.76 \pm 0.04}$ | $\mathbf{12.15 \pm 0.06}$ | $\mathbf{2.20 \pm 0.08}$ | $\mathbf{10.39 \pm 0.10}$ | $2.04 \pm 0.07$ | $\mathbf{9.77 \pm 0.10}$ |
| SIR-EBSW-e L=10 | $2.79 \pm 0.03$ | $12.26 \pm 0.05$ | $2.26 \pm 0.08$ | $10.53 \pm 0.09$ | $2.08 \pm 0.11$ | $9.87 \pm 0.16$ |
| IMH-EBSW-e L=10 | $2.82 \pm 0.02$ | $12.33 \pm 0.02$ | $2.26 \pm 0.12$ | $10.53 \pm 0.20$ | $2.07 \pm 0.02$ | $9.86 \pm 0.03$ |
| RMH-EBSW-e L=10 | $2.86 \pm 0.04$ | $12.37 \pm 0.03$ | $2.21 \pm 0.01$ | $10.45 \pm 0.05$ | $\mathbf{2.02 \pm 0.02}$ | $9.78 \pm 0.01$ |
| IS-EBSW-1 L=10 | $2.84 \pm 0.01$ | $12.43 \pm 0.01$ | $2.28 \pm 0.10$ | $10.63 \pm 0.11$ | $2.10 \pm 0.05$ | $9.91 \pm 0.05$ |
| SIR-EBSW-1 L=10 | $2.84 \pm 0.01$ | $12.38 \pm 0.01$ | $2.28 \pm 0.07$ | $10.59 \pm 0.10$ | $2.07 \pm 0.07$ | $9.88 \pm 0.12$ |
| IMH-EBSW-1 L=10 | $2.82 \pm 0.01$ | $12.36 \pm 0.03$ | $2.28 \pm 0.08$ | $10.52 \pm 0.05$ | $2.08 \pm 0.06$ | $9.86 \pm 0.09$ |
| RMH-EBSW-1 L=10 | $2.89 \pm 0.04$ | $12.47 \pm 0.03$ | $2.21 \pm 0.03$ | $10.45 \pm 0.08$ | $2.03 \pm 0.03$ | $9.80 \pm 0.02$ |

## C.3 Deep Point-cloud Reconstruction

We follow the same experimental settings as in the main text. We provide the full experimental results including the IS-EBSW, the SIR-EBSW, the IMH-EBSW, and the RMH-EBSW with both the exponential energy function and the identity energy function, with both $L = 10$ and $L = 100$ in Table 5. In Table 5, we use the conventional gradient estimator for the IS-EBSW while other variants of EBSW use the "parameter-copy" gradient estimator. We also compare gradient estimators for the EBSW by adding the results for the "parameter-copy" gradient estimator for the IS-EBSW (denoted as (c)), and the conventional gradient estimator for the SIR-EBSW, the IMH-EBSW, and the RMH-EBSW in Table 6.

**Quantitative Results.** From the two tables, we observe that the IS-EBSW-e performs the best for both settings of the number of projections $L = 10$ and $L = 100$ in terms of the Wasserstein-2 reconstruction errors. For the SW reconstruction error, it is only slightly worse than the SIR-EBSW-e at epoch 200. Comparing the exponential energy function and the identity energy function, we observe that the exponential function is better in both settings of the number of projections. For the same number of projections, the EBSW variants with both types of energy function give lower errors than the baseline including the SW, the Max-SW, and the v-DSW. For all sliced Wasserstein variants, a higher value of the number of projections gives better results. For the gradient estimator of the EBSW, we see that the conventional gradient estimator is preferred for the IS-EBSW while the "parameter-copy" estimator is preferred for other EBSW variants.

**Qualitative Results.** We show some ground-truth point-clouds ModelNet40 and their corresponding reconstructed point-clouds from different models ($L = 100$) at epochs 200 and 20 in Figure 6- 7 respectively. From the top to the bottom is the ground truth, the SW, the Max-SW, the v-DSW, the IS-EBSW-e, the SIR-EBSW-e, the IMH-EBSW-e, and the RMH-EBSW-e.

## C.4 Deep Generative Modeling

We follow the same setup as sliced Wasserstein deep generative models in [29]. We compare IS-EBSW-e with SW, and DSW [30] on CIFAR10 (image size 32x32) [21], and CelebA [23]. We show both FID score [15] (CIFAR10, CelebA) and IS score [39] (CIFAR10) in Figure 8. Overall, IS-EBSW-e performs better than SW and DSW in this task which suggests that EBSW has a wide range of applications.

Table 6: Reconstruction errors of different autoencoders measured by the (sliced) Wasserstein distance ($\times 100$). We use (c) for the "parameter-copy" gradient estimator. The results are from three different runs.

| Distance | Epoch 20 | | Epoch 100 | | Epoch 200 | |
|---|---|---|---|---|---|---|
| | $SW_2(\downarrow)$ | $W_2(\downarrow)$ | $SW_2(\downarrow)$ | $W_2(\downarrow)$ | $SW_2(\downarrow)$ | $W_2(\downarrow)$ |
| IS-EBSW-e L=100 (c) | $2.74 \pm 0.04$ | $12.14 \pm 0.12$ | $2.22 \pm 0.07$ | $10.42 \pm 0.05$ | $2.07 \pm 0.01$ | $9.77 \pm 0.07$ |
| IS-EBSW-1 L=100 (c) | $2.83 \pm 0.01$ | $12.34 \pm 0.03$ | $2.30 \pm 0.05$ | $10.60 \pm 0.09$ | $2.05 \pm 0.07$ | $9.83 \pm 0.11$ |
| SIR-EBSW-1 L=100 | $2.80 \pm 0.02$ | $12.29 \pm 0.01$ | $2.21 \pm 0.05$ | $10.46 \pm 0.08$ | $2.04 \pm 0.02$ | $9.81 \pm 0.07$ |
| IMH-EBSW-1 L=100 | $2.96 \pm 0.05$ | $12.67 \pm 0.08$ | $2.35 \pm 0.05$ | $10.82 \pm 0.07$ | $2.20 \pm 0.11$ | $10.20 \pm 0.16$ |
| RMH-EBSW-1 L=100 | $3.00 \pm 0.06$ | $12.67 \pm 0.10$ | $2.27 \pm 0.02$ | $10.66 \pm 0.06$ | $2.15 \pm 0.05$ | $10.11 \pm 0.11$ |
| IS-EBSW-e L=10 (c) | $2.77 \pm 0.01$ | $12.22 \pm 0.04$ | $2.28 \pm 0.09$ | $10.63 \pm 0.11$ | $2.07 \pm 0.07$ | $9.80 \pm 0.15$ |
| IS-EBSW-1 L=10 (c) | $2.86 \pm 0.02$ | $12.42 \pm 0.02$ | $2.24 \pm 0.08$ | $10.52 \pm 0.13$ | $2.05 \pm 0.04$ | $9.84 \pm 0.10$ |
| SIR-EBSW-1 L=10 | $2.87 \pm 0.02$ | $12.43 \pm 0.08$ | $2.36 \pm 0.11$ | $10.67 \pm 0.19$ | $2.08 \pm 0.10$ | $9.88 \pm 0.14$ |
| IMH-EBSW-1 L=10 | $2.98 \pm 0.02$ | $12.65 \pm 0.04$ | $2.35 \pm 0.05$ | $10.84 \pm 0.06$ | $2.21 \pm 0.11$ | $10.22 \pm 0.11$ |
| RMH-EBSW-1 L=10 | $3.01 \pm 0.04$ | $12.82 \pm 0.05$ | $2.37 \pm 0.03$ | $10.87 \pm 0.03$ | $2.11 \pm 0.02$ | $10.13 \pm 0.06$ |

# D   Computational Infrastructure

For the point-cloud gradient flows and the color transfer, we use a Macbook Pro M1 for conducting experiments. For deep point-cloud reconstruction, experiments are run on a single NVIDIA V100 GPU.

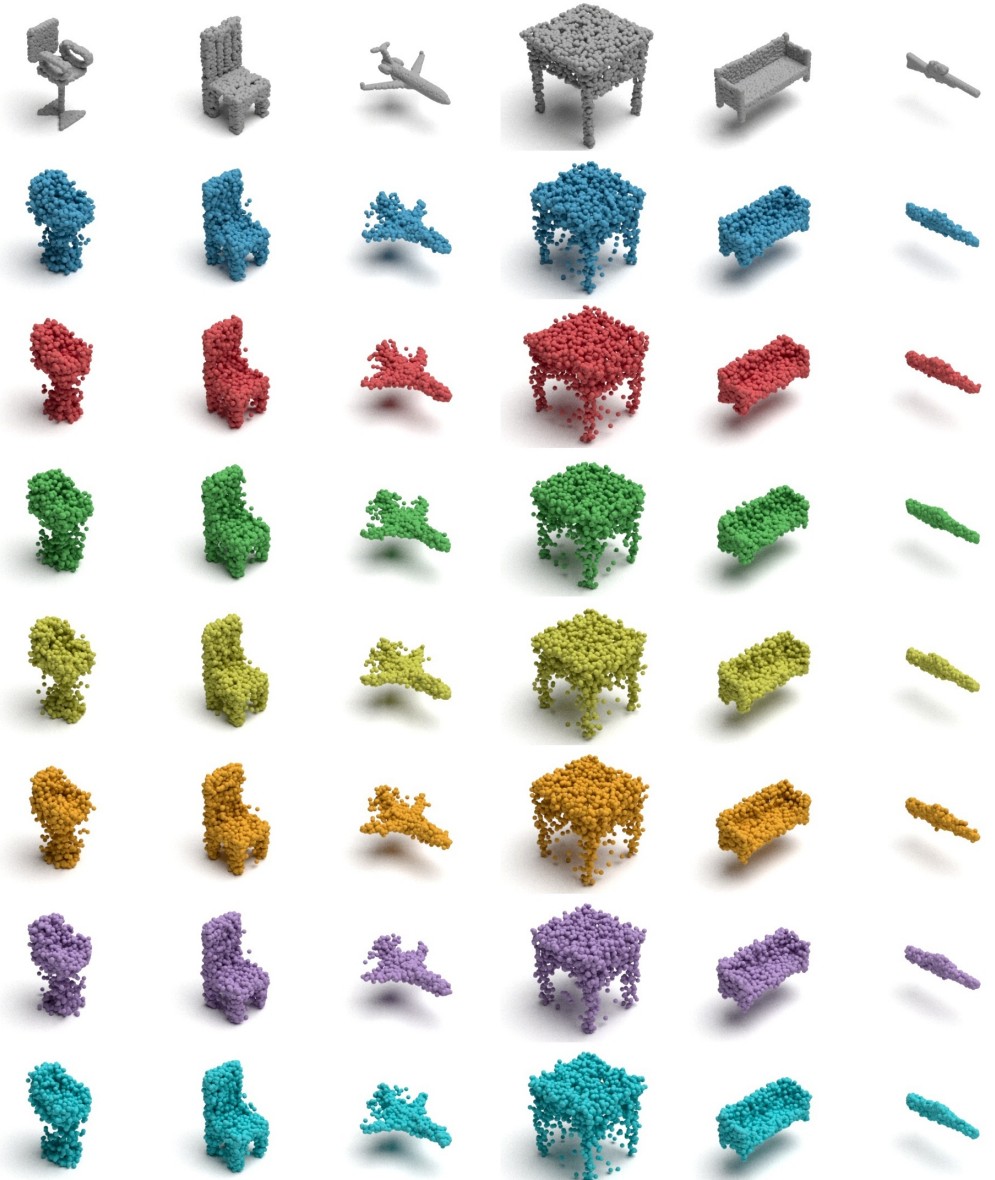

Figure 6: From the top to the bottom is the ground truth, the reconstructed point-clouds at epoch 200 of the SW, the Max-SW, the v-DSW, the IS-EBSW-e, the SIR-EBSW-e, the IMH-EBSW-e, and the RMH-EBSW-e respectively.

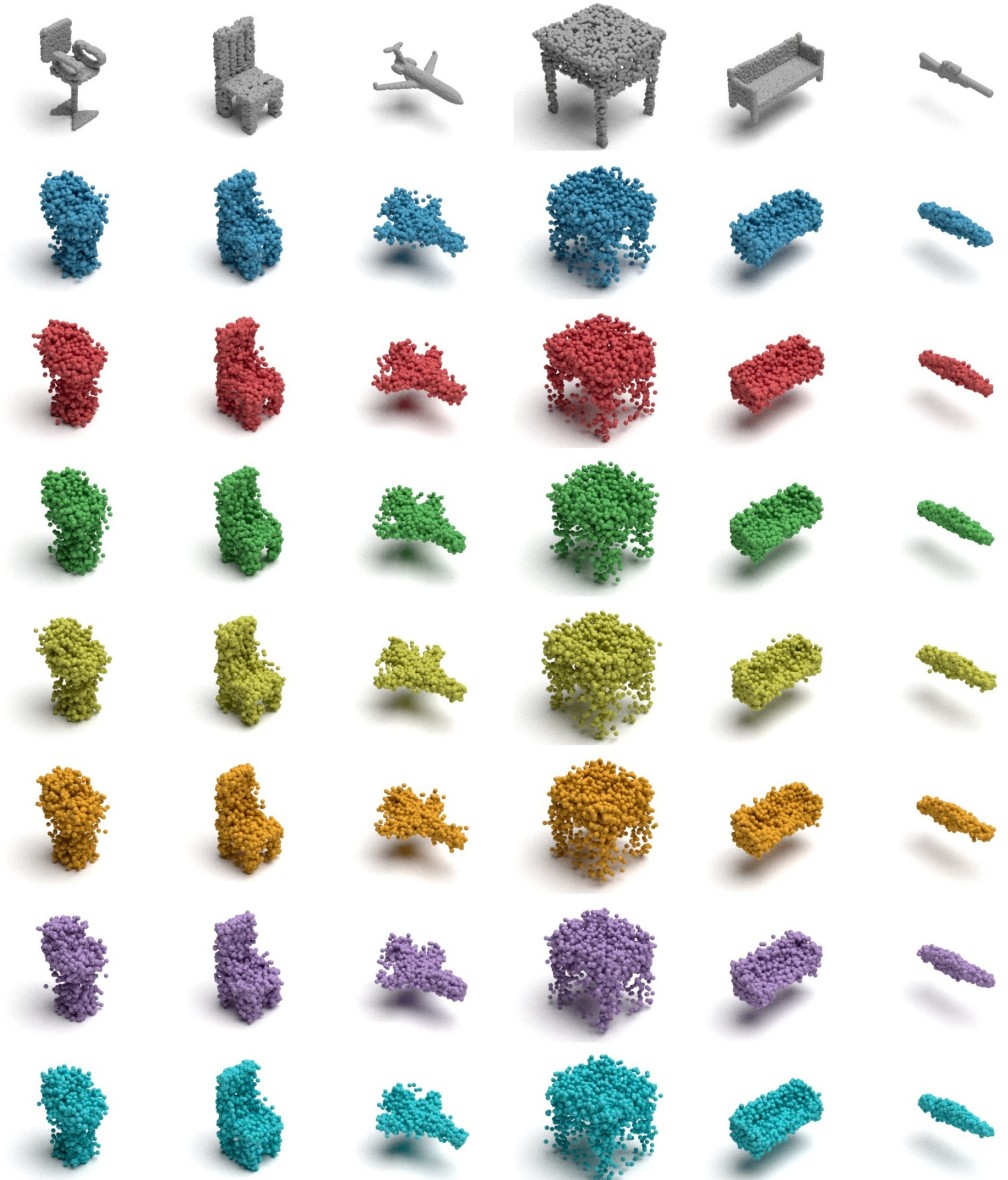

Figure 7: From the top to the bottom is the ground truth, the reconstructed point-clouds at epoch 20 of the SW, the Max-SW, the v-DSW, the IS-EBSW-e, the SIR-EBSW-e, the IMH-EBSW-e, and the RMH-EBSW-e respectively.

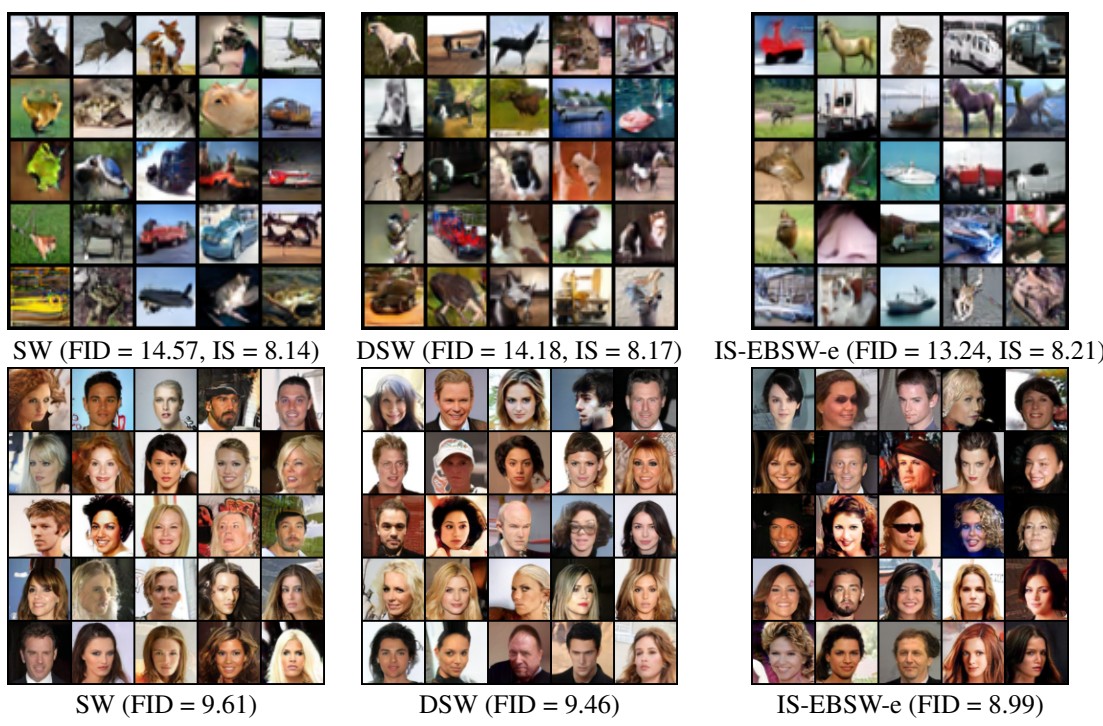

SW (FID = 14.57, IS = 8.14)     DSW (FID = 14.18, IS = 8.17)     IS-EBSW-e (FID = 13.24, IS = 8.21)

SW (FID = 9.61)       DSW (FID = 9.46)       IS-EBSW-e (FID = 8.99)

Figure 8: Generative modeling from SW, DSW, and IS-EBSW-e with L=100.

