# OpenReview forum: "Energy-Based Sliced Wasserstein Distance"
_NeurIPS.cc/2023/Conference — NeurIPS 2023 poster_

### Official Review · Reviewer_krQV · 2023-07-03

**Soundness:** 3 good
**Presentation:** 3 good
**Contribution:** 3 good
**Rating:** 6
**Confidence:** 4

**Summary:**

This paper introduces a new distribution of slices of the Sliced-Wasserstein distance based on the energy-based model framework. The authors study the theoretical properties and conduct numerical experiments on EBSW.

**Strengths:**

- The study of the idea is organized and clear.
- Theoretical quantities of interest are derived.
- Various sampling methods are provided.

**Weaknesses:**

- The method is not compared with the vanilla Distributional Sliced-Wasserstein method which should be the baseline to beat. The fact that only the vMF-DSW method is benchmarked is a weakness, as the learned distribution is unimodal, which prevents the model to be too expressive.
- I think it would be good to have a theorem studying the following sample complexity, that is more of interest for practical reasons:
$ \mathbb{E}[EBSW_p(\mu_n,\nu_n;f) - EBSW_p(\mu,\nu;f)]$
- Experiments are a bit light, I think a generative modeling experiment would be nice to have as this is one of the main applications of the SW distances.

**Questions:**

- Can the authors explain why experimental results are not compared to the DSW distance?
- It appears to me that EBSW falls into the family of adaptive Sliced-Wasserstein distances ( https://arxiv.org/pdf/2206.03230.pdf , which doesn't seem to be cited in the related work section). Can the authors comment on that and whether this framework can help studying this distance?

**Limitations:**

None. I am willing to discuss with the authors and accordingly modify my score.

---

> ### Author Rebuttal · Authors · 2023-08-08
>
> We appreciate the review's feedback and comments and want to express our thanks. Our responses are as follows. We remain open to additional questions and further discussion.
>
> **Q19**: The method is not compared with the vanilla Distributional Sliced-Wasserstein method which should be the baseline to beat. The fact that only the vMF-DSW method is benchmarked is a weakness, as the learned distribution is unimodal, which prevents the model to be too expressive.
>
> **A19**:  We would like to refer the reviewer to the global rebuttal for additional experiments comparing our method with vanilla DSW. In the submission. the primary reason we chose the von Mises-Fisher (vMF) distribution as the family of slicing distributions is its simplicity and parameter efficiency. The vMF distribution has only two parameters: the location parameter (a d-dimensional vector) and the concentration parameter (a scalar). Therefore, performing stochastic optimization for vMF is expected to be more stable. In contrast, using an overparameterized family of distributions, such as the push-forward distribution with neural networks in vanilla DSW, can be computationally expensive. For instance, using an MLP neural network could increase the computational complexity to be proportional to $d^2$ due to matrix multiplications, as opposed to the original scaling of $d$ in SW and EBSW. Additionally, vanilla DSW requires a constraint to control the concentration of the push-forward distribution, making the optimization less interpretable and potentially more challenging. Moreover, solving vanilla DSW necessitates an admissible regularizing constant and involves using duality, which could further complicate the optimization problem.
>
> It's important to note that we are not challenging DSW in its population form but rather in its computational form. Utilizing a powerful family of slicing distributions with a high number of parameters would incur increased computational time, memory usage, and hyperparameter tuning. In our paper, we present a budget-constrained comparison, specifically the scaling constraint of $n \log n$. Consequently, we believe that using vanilla DSW may not offer significant improvements over vDSW. We would like to emphasize the main advantage of EBSW, which lies in its simplicity of use (choosing an energy function is easier than selecting a family of distributions) and computational efficiency (parameter-free and optimization-free).
>
> **Q20**: On two-sided sample complexity
>
> **A20**: Thank you for your insightful question. Proposition 1 can be referred to as the one-sided sample complexity, which is useful in certain settings, e.g., quantization of a distribution (clustering). The sample complexity you mentioned is called the two-sided sample complexity. Normally, we can directly derive the two-sided sample complexity from the one-sided sample complexity by using the triangle inequality. Unfortunately, the triangle inequality of EBSW has not been proven in our paper, due to the expressiveness of the energy-based slicing distribution. Specifically, while we can prove the triangle inequality using the triangle inequality of Wasserstein distance and the Minkowski inequality for a fixed slicing distribution, it is not trivial to establish this inequality for the adaptive slicing distribution of EBSW. We believe that proving the triangle inequality for EBSW presents a challenging theoretical problem, and we leave this open problem for future work.
>
>
> **Q21**: Experiments are a bit light, I think a generative modeling experiment would be nice to have as this is one of the main applications of the SW distances.
>
> **A21**: We would like to mention that it takes about one day to complete the training of a deep point-cloud autoencoder on a Tesla V100 GPU. Therefore, we believe our experiments are relatively intensive compared to the current literature on proposing new SW variants [1] [2]. Nevertheless, we agree that generative modeling is one of the main applications of SW distances. For detailed additional experiments on deep generative modeling, we refer the reviewer to the global rebuttal.
>
>
> [1] Generalized sliced Wasserstein distances, Kolouri et al
>
> [2] Spherical Sliced Wasserstein, Bonet et al
>
>
> **Q22**: Can the authors explain why experimental results are not compared to the DSW distance?
>
>
> **A22**: We do use v-DSW in the paper as an instance of DSW, which is more interpretable and simpler. As discussed in **Q21**, vanilla DSW requires more parameters and computation compared to vDSW due to the usage of neural networks for push-forward slicing distribution. However, we have added a comparison with DSW in the rebuttal PDF which shows the favorable performance of EBSW. We would like to refer the reviewer to the global rebuttal for detailed experimental results.
>
>
> **Q23**: It appears to me that EBSW falls into the family of adaptive Sliced-Wasserstein distances ( https://arxiv.org/pdf/2206.03230.pdf , which doesn't seem to be cited in the related work section). Can the authors comment on that and whether this framework can help studying this distance?
>
>
> **A23**; Thank you for providing the reference to the new ICML 2023 paper. We will include this paper in our references and discussions. In the paper, they define adaptive sliced Wasserstein distances (ASW) based on an optimization problem, similar to DSW. In contrast, EBSW is an optimization-free variant of SW. EBSW employs an adaptive slicing distribution, which is explicitly constructed, unlike the implicit construction in ASW and DSW. It would be interesting to explore if we can formulate EBSW as the optimal solution for ASW and DSW, under certain choices in the family of slicing distributions and regularity conditions. With such a connection, we can leverage the theoretical results from the mentioned paper to study EBSW in greater detail. Overall, we believe this direction holds great promise, and we will leave this investigation for future research.

---

> ### Author Response · Authors · 2023-08-22
> **Thank You**
>
> Dear Reviewer krQV,
>
> Thanks for your reviews of our paper. Since the discussion period between the authors and the reviewers was already over and we have not heard from you during this period, we would be grateful if the reviewer could let us know if all your questions are addressed to some extent. If you are satisfied with our answers, we hope that the reviewer will consider adjusting your score.
>
> Best,
>
> The Authors

---

> > ### Comment · Reviewer_krQV · 2023-08-22
> > **Answer to rebuttal**
> >
> > Dear reviewers, I would like to apologize for the late answer and to thank you for the rebuttal.
> > I am happy with the comparison with DSW that I hope will be added to the paper.
> >
> > I have the feeling that a little bit more effort should have been put into proving the triangle inequality for EBSW and the two-sided sample complexity even though I don't know how hard it is to prove.
> >
> > I am however raising my score from 5 to 6 as I think the authors did a good rebuttal.

---

### Official Review · Reviewer_vaua · 2023-07-06

**Soundness:** 3 good
**Presentation:** 3 good
**Contribution:** 3 good
**Rating:** 6
**Confidence:** 4

**Summary:**

This paper proposes an extension to Sliced Wasserstein Distance (SW), an approach for measuring distances between distributions by computing the average of the energy of the 1-d Wasserstein distances between 1-d projections. The authors argue that moving towards non-uniform ways of sampling the projections is key, and subsequently show that the benchmark approach currently that does so (Distributional sliced Wasserstein) adds another optimization over a parametric distribution family, which may not yield stable results and is more computationally expensive. Subsequently, they propose a simple non-parametric extension to SW called Energy based SW (EBSW) that achieves the same objective of sampling non-uniformly, i.e. sampling projections with larger Wasserstein distance with a greater probability, by considering energy functions $f$ which are monotonically increasing w.r.t the sliced 1-d Wasserstein distance itself. Theoretical properties are highlighted, including the convergence properties of their proposed metric. Next, the authors propose a range of Monte Carlo estimation methods to approximate EBSW, including deriving the necessary gradient expressions to be used in some of their subsequent applications. Lastly, over multiple experiments ranging from point-cloud gradient flows to deep point-cloud reconstruction, the authors demonstrate that EBSW is faster (Importance Sampling approach), and yields distributions closer to the ground truth in most cases than other SW-based benchmarks.


**Strengths:**

I really liked reading the paper and going through the various findings in it. The following are what I consider to be some of the main strengths of this work. (i) The paper is really well-written. All sections were intuitive to read and easy to understand for me, and this also applied to the Appendices, including the proofs of the Theorems and Propositions. (ii) The contributions of the paper are very clear and supported by the theoretical and empirical results. All proposed benefits of the approach have been verified by the authors in theory and practice. (iii)  The experiments are well motivated and relatively exhaustive. The proposed approach is tested in a wide range of problems and the results convincingly showcase the benefits of EBSW. (iv) The background section is well organized and is an effective introduction to SW-based metrics in general.


**Weaknesses:**

The paper is overall interesting and well-written. However, there are a few points noted below that can improve the work further.
1. I feel that the theoretical results can be given a bit more perspective w.r.t. the other SW-based metrics. Overall, I felt the theoretical results in Theorems 1, 2 and Propositions 1 and 2 are relatively intuitive to show. Some more specific insights on EBSW could be interesting (perhaps with constrained energy functions, i.e. Lipschitz constrained $f$).
2. I understand the overall direction for moving towards measures that look disproportionately at projections which have larger distances (lines 37-38). However, I feel that a few more words to elaborate on intuitively why one must move towards non-uniform distributions in the Sliced Wasserstein setting could be useful.
3. One of the main questions I have is the role of $p$, which is fixed for all experiments in this work. I feel that the authors can elaborate a bit on whether trying larger values of $p$ can achieve a similar goal to what EBSW-e achieves (More details in questions).
4. The result tables are convincing and informative w.r.t. the tested methods. However, in some of the visual depictions, I found it hard to see any significant differences between EBSW and some other variants (even in the Appendices). Mainly, the color transfer experiments were a bit hard to discern for me in terms of relative performance

**Questions:**

In addition to the list of weaknesses above, below are some questions and general suggestions for the authors to address:
1. I noticed that the authors fix the value of $p$ to 2 for all experiments. I wonder if increasing $p$ can achieve a similar effect to using a monotonically increasing energy function (such as the exponential distribution in this work). Because intuitively, it seems to me that increasing $p$ eventually also gradually assigns more importance to the projections with values near to the maximum. I’m mainly asking this because increasing $p$ still falls under SW-p, and thus could perhaps have similar sample complexity and computational load. Could you please provide a theoretical explanation on how changing $p$ fundamentally differs from EBSW with an increasing energy function?
2. For Figure 2, the authors state that the gradient flows for EBSW are smoother than other approaches. Perhaps describing some visual cues to elaborate on that point can make it easier to follow.
3. Although four different ways of estimating the EBSW variants are proposed, I only see the IS-EBSW being reported in the main paper. Furthermore, having looked at the results in the appendix, it seems in almost all cases, the variants have roughly similar performance, and IS-EBSW seems to lead to better results in most cases. Furthermore, it also seems that the time complexity for all algorithms are similar, also seeing Table 3 in C.1, it seems that in most cases IS-EBSW is faster as well. Thus, my question is: in discussing these other approaches, are there benefits to the other proposed MCMC variants?
4. Are there any cases where the choice of an exponential energy function $f$ doesn't work? It would be intuitive to me that, considering a more general family of exponentials in $e^{(\lambda x)}$, the optimal "width" of the slicing distribution as set by $\lambda$ may be different for different cases. For instance, can there be any cases where the underlying distribution enforces a much larger or smaller $\lambda$ than one? Or is $\lambda=1$ in some sense universal?
5. As a follow-up to the previous question, I would assume that for the optimization results in Tables 1 and 2, as the generated distribution edges closer to the ground truth distribution, would it be profitable to increase $\lambda$ higher than one? As then the distances would all get very small, and potentially max-SW would start to look more appealing.
6. The theoretical results are interesting and important, however, could the authors put into perspective how the theoretical properties of EBSW compare to other SW based measures? It seems to me that both Theorems 1 and 2 may hold for SW based measures (at least SW, not sure about max-SW and DSW). Similarly, for Proposition 2 it seems from the proof that it would hold for both SW and max-SW (not sure about DSW). Additional theoretical results in this direction, or at the least a rough intuitive discussion on how these results would compare for the other SW counterparts, would be useful. This would put more into perspective how EBSW's theoretical properties compare to the other SW benchmarks.

**Limitations:**

Some limitations have been discussed, but I feel there is scope to discuss more limitations of the proposed approach.

---

> ### Author Rebuttal · Authors · 2023-08-08
>
> We'd like to express our gratitude for the constructive feedback. We're here to provide responses as outlined below. We are eager to participate in further discussions.
>
> **Q10**: On Lipschitz constrained $f$
>
> **A10**: Regarding the Lipschitz energy function, the polynomial function is Lipschitz and the exponential function is locally Lipschitz. Lipschitzness could lead to a potential notion of robustness to the distance. Specifically, when two measures are contaminated with additive noise, a significant change in the Wasserstein value from two closed projecting directions could be due to the noise. Therefore, using a Lipschitz energy function could penalize the weight given to such projecting directions and has benefits.
>
> From a Lipschitz energy function, we can show that the energy-based slicing distribution has a Lipschitz probability density function (pdf) since the SW distance is Lipschitz with respect to the projecting direction [1]. However, an exact computation of the Lipschitz constant is not trivial in the case of the SW distance.
>
> [1] Statistical, Robustness, and Computational Guarantees for Sliced Wasserstein Distances, Goldfeld et al.
>
> **Q11**: Why one must move towards non-uniform distributions in the  SW setting could be useful?
>
> **A11**: It benefits applications that have a sequence of measures that converge to a target measure e.g., gradient flow, and generative modeling. When using SW as the criteria to drive the sequence, the current measure is derived toward the target measure from all directions equally. This is undesirable since two measures can be closed in some directions and can be significantly different in other directions. Hence, the uniform distribution leads to sequences that converge slower.
>
> **Q12**:whether trying larger values of $p$ can achieve a similar goal to what EBSW-e achieves? How changing  $p$  fundamentally differs from EBSW?
>
> **A12**:  As in Proposition 1, we show that EBSW is an upper bound of SW for any $p$.  Adjusting $p$ equals to change the notation of ground metric on the support sets of two measures. However, changing $p$ in SW does not affect the slicing distribution which is still uniform. Therefore, larger values of $p$ cannot achieve a similar goal as EBSW since EBSW has a non-uniform distribution. To verify further the intuition, we run the experiments on gradient flow again for SW with p=3 and p=10 in the global rebuttal.
>
> In contrast, changing $p$ does affect the EBSW  since the energy distribution is proportional to the composition between $f$ and $L_p$ norm. Due to the flexibility in choosing $f$, the interaction between $f$ and the norm is expressive and worth a careful future investigation.
>
> **Q13**:  On visual comparison in GF.
>
> **A13**: One way to see the difference from our experience is by looking at the level of “yellow” and “orange" colors. In more detail, the level of yellow color is more than the level of orange color in the target image while the reverse phenomenon happens in the source image. From that perspective, the image from EBSW has more yellow and less orange than images from other distances.
>
> **Q14**: On some visual cues to elaborate on the performance of EBSW in gradient flow.
>
> **A14**: From the figure, we observe that the legs of the table  (the source point-cloud) are bent in SW, Max-SW, and vDSW in the first step and do not look like a real table. In contrast, EBSW can create a more real table with a sharp connection between the legs and the desktop, and the legs are also not bent. In future steps, we see that the point-clouds from EBSW have fewer jumping points on the surface e.g., the cover of the light bulbs.
>
> **Q15**: benefits of proposed MCMC variants?
>
> **A15**: As discussed in the global rebuttal, MCMC variants are slow since they are not parallelable.  However,  using MCMC variants could benefit the approximation when we have a very peaky slicing density. In that case, important sampling might be not accurate since the proposal distribution could not yield good samples due to the concentrated mass of the slicing density. Moreover,  the importance sampling is also unstable to use due to the huge value of the density ratio between the slicing density and the proposal density. However, MCMC can overcome these issues with a good transition distribution and achieve a better approximation.
>
> **Q16**:  On the energy function $e^{\lambda x}$... Would it be profitable to increase $\lambda>1$?
>
> **A16**: With the exponential energy function, the importance sampling reverts to using the Softmax function. For your suggestion, it leads to the annealed exponential function, we get the annealed Softmax function which is better at controlling the energy density. However, it costs one hyperparameter in addition. A potentially useful choice of $\lambda$ is $1/p$ which helps us to prevent the effect of changing $p$ to the slicing distribution as discussed in **Q12**.
>
> Increasing $\lambda$ leads to a peaker slicing density and makes the density closer to the Dirac delta at the max slice. However, a peaky density is hard and unstable to approximate as discussed in **Q15**. Secondly, such kind of peak density is not desirable since they are strongly affected by the noise as discussed in **Q10**.
>
> **Q17**: How EBSW's theoretical properties compare to the other SWs?
>
> **A17**: Theorem 2, and Proposition 2 mean that EBSW retains nice properties as SW and Max-SW i.e., weak convergence, and no curse of dimensionality. However, in Theorem 1, the triangle inequality of EBSW has not been proved yet due to the adaptive slicing distribution. Given a fixed slicing distribution, the triangle inequality can be proved using the metricity of Wasserstein distance and Minkowski inequality. In contrast, it is not trivial how to obtain the triangle inequality with the energy-based slicing distribution in EBSW.
>
> **Q18**: On Limitation
>
> **A18**: We would like to refer the reviewer to the global rebuttal of the limitations of our work.

---

> > ### Comment · Reviewer_vaua · 2023-08-21
> >
> > I thank the authors for their detailed replies to my questions and comments. I am happy to keep my rating. Thanks again!

---

> > > ### Author Response · Authors · 2023-08-21
> > > **Response to Reviewer**
> > >
> > > We want to thank the reviewer for keeping the score at 6. We will include our discussion in the revision of the paper.
> > >
> > > Best regards,

---

### Official Review · Reviewer_dYij · 2023-07-07

**Soundness:** 3 good
**Presentation:** 2 fair
**Contribution:** 3 good
**Rating:** 6
**Confidence:** 3

**Summary:**

This paper has proposed an energy-based slicing distribution that maps original distributions into one-dimesion space  to compute the Wasserstein distance.

**Strengths:**

This paper has proposed an energy-based slicing distribution that maps original distributions into one-dimesion space  to compute the Wasserstein distance. The proposed method shows great advantages over other slicing distribution like sliced Wasserstein, Distributional sliced Wasserstein, and Max sliced Wasserstein. The authors also explore different sampling methods to approximate the value of the EBSW distance.

**Weaknesses:**

1. An introduction about optim transport should be included in the Background section.
2. Put a title for each row in Figure 2 could help reader easier to compare different SW methods.

**Questions:**

1. In definitition 1, does $\theta$ represent parameters in the slicing distribution or just the transport function? If  $\theta$ are parameters, why could the model be called as parameter-free?

2. In general, the energy function should be defined as $f:(-\inf, \inf)^d \rightarrow [0, \inf]$. Why the author defines it as in line 144?

3. Could the proposed distance measurement be used in generative models, like GANs and EBMs?

4. Is there any comparison with other probability distance measurement methods, like KL divergence, in the experiments?




**Limitations:**

This paper has proposed an energy-based slice distribution to compute Wasserstein distance between two distributions. My question mainly lies in the application of energy function and comparison of methods based on other  probability measurement methods.

---

> ### Author Rebuttal · Authors · 2023-08-07
>
> We begin by expressing our gratitude for the insightful feedback and comments provided in the review. In response, we offer the following explanations and answers to your inquiries. We welcome any further questions and are open to engaging in a more in-depth discussion on the matter.
>
> **Q5**: An introduction about optimal transport should be included in the Background section.
>
> **A5**: Thank you for your suggestion. Due to the space constraint of the main paper, we skip the definition of optimal transport and Wasserstein distance. From your feedback, we realize that it affects the readability of the paper. Therefore, we will add those backgrounds to the paper in the revision.
>
> **Q6**: Put a title for each row in Figure 2 could help reader easier to compare different SW methods.
>
> **A6**: Thank you for your suggestions. We will add the legend to the figure for better readability.
>
> **Q7**: In definition 1, does $\theta$ represent parameters in the slicing distribution or just the transport function? If $\theta$  are parameters, why could the model be called as parameter-free?
>
> **A7**: Thank you for your insightful question. In Definition 1, $\theta$ is the realization of the distribution, not the parameter. For a more detailed explanation, in DSW, $\theta \sim vMF(\epsilon,\kappa)$, where $\epsilon$ and $\kappa$ are parameters, while EBSW does not have such parameters. Therefore, EBSW is parameter-free. To elaborate further, EBSW is defined based on the energy function which non-parametric and is easier and more flexible to choose than choosing a family of distributions as in DSW. In addition, in its computational form, Max-SW and DSW require additional hyperparameters for optimization, such as learning rate, the number of gradient steps, optimizers, and so on, while EBSW has only one hyperparameter, which is the number of projections, like the vanilla SW. Overall, this is the reason we refer to EBSW as parameter-free and optimization-free. We will include this explanation in the revision.
>
> **Q8**: In general, the energy function should be defined as  $f :(−\infty,\infty)^d\to [0,\infty]$. Why the author defines it as in line 144?
>
> A8: Thank you for your question.  We would like to clarify it as follow. The energy function takes the value of the $p$-power Wasserstein distance between two projected one-dimensional measures and yields a positive number. Since the Wasserstein distance is a scalar and always positive, the pre-image of the energy function is in the range of $[0, \infty]$. Based on the energy function, we can define the energy-based slicing distribution, which is a distribution on the unit-hypersphere and has a probability density function (pdf): $\mathbb{S}^{d-1} \to \Theta \in (0, \infty)$, as described in Definition 1.  The reason we avoid having the point 0 in the image of the energy function is to prevent the slicing distribution from being undefined when two measures coincide. This ensures the distribution's well-defined nature and allows us to use it safely in practical applications. We will add a more detailed explanation to the revision.
>
>
> **Q8**: Could the proposed distance measurement be used in generative models, like GANs and EBMs?
>
> **A8**: The answer is yes. The reason we focus on point-cloud applications since they are disjointed supports that other divergences such as KL divergence, Jensen Shannon divergence, f-divergence cannot handle. EBSW can absolutely be used in generative modeling. We would like to refer the reviewer to the global rebuttal for a detailed additional result on deep generative modeling. Overall, IS-EBSW-e has shown a favorable performance compared to SW and DSW in generative modeling. However, we suggest using this additional result as evidence to show that EBSW is versatile since we only a limited time in the rebuttal period.
>
> **Q9**: Is there any comparison with other probability distance measurement methods, like KL divergence, in the experiments?
>
> **A9**: As stated in **Q8**, the KL divergence (one member of f-divergence) is not suitable for handling disjointed support measures. The reason behind this is that the KL divergence relies on the density ratio, which becomes undefined when the denominator is equal to 0. Consequently, KL divergence cannot be directly applied to compare two-point clouds. If one wishes to use KL divergence between two point clouds, it becomes necessary to convert them into voxels (3D bins). Nevertheless, this transformation is not differentiable, rendering it unsuitable for deep learning applications involving point clouds. Therefore, using KL divergence in such scenarios is not practical.
> Moreover, it is important to note that KL divergence is not symmetric, while Wasserstein variants are all symmetric. This means that the Wasserstein distance and its variants, such as the Earth Mover's Distance, do not depend on the ordering of the points and provide a more symmetrical measure for comparing two probability distributions.
>
> For a comparison in generative modeling, training conventional SNGAN can be seen as an application of Jensen Shannon divergence (which can be seen as a symmetric version of KL)  which achieves a FID score of   21 [1]. By using the same architecture, we achieve a FID score of 13.24 with IS-EBSW-e in the rebuttal pdf.
>
> [1] Spectral Normalization for Generative Adversarial Networks, Miyato et al
>
> In summary, due to its limitations with disjointed support measures and its lack of symmetry, KL divergence is not the most appropriate choice for comparing point clouds directly. Wasserstein-based distance metrics offer more suitable alternatives for handling these scenarios, especially in deep learning applications involving point clouds.

---

> > ### Comment · Reviewer_dYij · 2023-08-19
> >
> > Thanks for the feedback from the authors. I will remain my rating.

---

> > > ### Author Response · Authors · 2023-08-19
> > > **Response to reviewer**
> > >
> > > We would like to thank the reviewer for keeping a positive score of 6. We are happy to discuss more if the reviewer still has questions.
> > >
> > > Best regards,
> > >
> > > Authors

---

### Official Review · Reviewer_QgRs · 2023-07-10

**Soundness:** 3 good
**Presentation:** 3 good
**Contribution:** 2 fair
**Rating:** 5
**Confidence:** 3

**Summary:**

This paper proposed a new variant of sliced Wasserstein distance that is inspired from Energy-Based Model called Energy Based Wasserstein Distance (EBWD). The proposed method models the energy function of the distance, from which the slices can be sampled. Three sampling techniques are proposed. The paper evaluate the proposed distance calculation using 3 tasks, Point Cloud Gradient flow, color transferring, and Point Cloud reconstruction. The results have shown that the new variant helps the training process convert faster and the transition between two distribution is smoother (in term of Wasserstein distance and Sliced Wasserstein distance getting small faster).

**Strengths:**

The paper has the following strengths

- A clear motivation to use the energy function. Similar to SW, the proposed energy-based SW enjoys a non-optimization based computation via sampling using the energy function.
- A detailed theoretical analysis of the proposed distance, accompanied by a nice experimental setup to evaluate its performance.

**Weaknesses:**

In general, I like the simple, yet interesting proposal from the paper. However, I also have a few concerns:

- It would be reasonable to evaluate EBSW in a more complex task such as image generation, similar to that where DSW is introduced. It seems like only IS-EBSW enjoys the low-computation benefit but other EBSW variants do not.
- It is not clear to me why EBSW is better than DSW while DSW finds the most separating directions. Especially the fact that EBSW converges faster.

**Questions:**

Please see comments in Weaknesses

**Limitations:**

The paper does not include a Limitation discussion.

---

> ### Author Rebuttal · Authors · 2023-08-07
>
> We wish to extend our appreciation for the valuable feedback and comments provided in the review. In response, we would like to address your questions as follows. We are readily available to address additional inquiries and to engage in further discussion.
>
> **Q1**: It would be reasonable to evaluate EBSW in a more complex task such as image generation, similar to that where DSW is introduced.
>
> **A1**: We acknowledge the importance of image generation as a task that sliced Wasserstein variants can excel at. In our paper, we did not include generative modeling due to the well-known instability associated with training generative models using mini-batch stochastic training algorithms. Instead, we focused on a stable setting for comparison and demonstrated the benefits of EBSW in the gradient flow application, which can be seen as a simplified and stable version of generative modeling. Nonetheless, we appreciate the reviewer's feedback and have taken it into consideration. To address this concern, we have now added the results of generative modeling to our paper. This addition showcases EBSW's applicability to various tasks. For detailed experiments and results, we refer the reviewer to the global rebuttal section, where we provide an overview of the experiments and outcomes. Overall, EBSW still shows a favorable performance compared to SW and DSW in this task. Nevertheless, due to the limited time of the rebuttal period, we suggest using the generative modeling result only as a reference for the versatility of EBSW.
>
>
> **Q2**:  It seems like only IS-EBSW enjoys the low-computation benefit but other EBSW variants do not.
>
> **A2**: Thank you for your insightful comment. The reason IS-EBSW is fast is that its computational algorithm is fully vectorizable, i.e., we can stack the supports of measures into matrices, stack the projecting directions into matrices, and then perform computations on matrices (e.g., matrix multiplication) that are parallelizable.
>
> For sampling importance resampling (SIR-EBSW), the key bottleneck is the resampling step. In particular, our implemented sampling algorithm is not parallel since the computation library (PyTorch) does not seem to support it to the best of our knowledge.
>
> Regarding Markov chain Monte Carlo (IMH-EBSW and RMH-EBSW), the sampling algorithms are naturally sequential; hence, they are not parallelizable. To the best of our knowledge, we could use multiple parallelizable Markov chains as a solution. However, that kind of speeding-up method needs a careful investigation of the mixing time of the MCMC algorithm to design the number of chains and the number of time steps for keeping a budget-constrained algorithm.
>
> However, we would like to recall that all discussed specific computation algorithms (IS, SIR, MCMC) in the paper are just basic variants in their vast literature and can be significantly improved. Therefore, we will leave this investigation for future work since our main focus of the paper is the energy-based slicing distribution and EBSW. At the moment, we recommend importance sampling (IS-EBSW) as the main computation method for EBSW due to its fast computation and simplicity.
>
>
> **Q3**: It is not clear to me why EBSW is better than DSW while DSW finds the most separating directions. Especially the fact that EBSW converges faster.
>
> **A3**: We would like to direct the reviewer to the global rebuttal concerning the discussion on DSW. In our paper, we present a budget-constrained comparison that places a computational time limit on distances, defined by a scaling constraint of $n \log n$. As a result, DSW might not attain optimality. Furthermore, an optimal design for the slicing distribution in DSW remains unknown. In our study, we employ the von Mises Fisher family for the slicing distribution, chosen for its simplicity and efficiency. Nevertheless, it is still slower than EBSW with importance sampling. In our additional experiment, we also use DSW with an implicit slicing distribution induced by a neural network; however, it still does not perform as effectively as IS-EBSW-e in applications. To clarify, we are not questioning DSW in its population form, but rather its practical usage, which presents challenges such as sub-optimality and high computational complexity. In conclusion, EBSW with importance sampling emerges as the preferred choice due to its simplicity, effectiveness, and efficiency.
>
> **Q4**: The paper does not include a Limitation discussion.
>
> **A4**: Thank you for pointing this out. In the revision, we have gathered all limitations of the papers into a new paragraph. We would like to refer the reviewer to the global rebuttal for the discussion about the limitation of the paper. Here, we would like to summarize the discussion. As discussed in **Q2**, one limitation of EBSW is its non-parallelizable MCMC variations, resulting in sluggish computations.  Furthermore, selecting suitable energy-based functions presents a challenge in applications. Finally, demonstrating the triangle inequality of EBSW is difficult due to the complexity of the energy-based slicing distribution. Overall, as discussed in the global rebuttal, these limitations are specific to the current paper itself, rather than EBSW.

---

> > ### Comment · Reviewer_QgRs · 2023-08-21
> > **Thank you for the responses!**
> >
> > Thank you for the detailed responses. Please see my additional comments:
> >
> > Q1. Thank you for the additional results on image generation. I notice that the FID of DSW is significantly better than that reported in the original DSW's paper. Also, all the methods have quite similar FIDs, which is difficult to make a conclusion as mentioned in the rebuttal.
> >
> > Q3. It's quite unfair to report the performance of DSW when it does not reach optimality. I think it is ok to report the performance of DSW even if it is better, although it incurs more computation. For example, one can trade-off between different estimation approaches, depending on how much computational budget they have. Currently, the experimental results seem to favor the proposed method also in quantitative metrics as well, which can be misleading.
> >
> > For the other questions, thank you for the clarifications. I will consider all the responses in my final rating of the paper.

---

> > > ### Author Response · Authors · 2023-08-21
> > > **Response to Reviewer**
> > >
> > > Thank you for your reply,
> > >
> > > On Q1: The reason the FID score is better since we utilize a stronger backbone for the generative models i.e., ResNet50. As mentioned in the global rebuttal, the main aim of the additional generative modeling experiments is to show the ability to apply widely of EBSW.
> > >
> > > On Q2: It is worth noting that we do not try to report unfairly for DSW. It is hard to know when DSW will reach its optimality and it could take a lot of computation. Due to the time limitation of the rebuttal, we must focus on the budget-constraint setting where we fix the budget of computation for all baselines. Since there are only 2 hours left of the discussion period, we cannot run additional experiments. However, we will add experiments on letting DSW have a large number of optimization updates in the revision. In the paper, we mainly focus on the computational aspect since EBSW is motivated by the computational limitation of previous variants. We will try to highlight this focus further in the revision.
> > >
> > > Since the paper is borderline now, we would be grateful if the reviewer could increase the score if all questions are addressed to some extent. We are happy to discuss more if the reviewer is not satisfied with our answers.
> > >
> > > Best regards,

---

> > > > ### Comment · Reviewer_QgRs · 2023-08-21
> > > > **Response to the authors!**
> > > >
> > > > Thank you for the responses to my questions. It is not my intention to ask for more experiments. The existing evaluation of the paper is overemphasizing the advantages of the proposed method, with limited guidance to its limitations. However, I think the proposed method is interesting and I will consider the responses in its final rating.

---

> > > > > ### Author Response · Authors · 2023-08-21
> > > > > **Response to Reviewer**
> > > > >
> > > > > Thank you for your quick reply.
> > > > >
> > > > > We are happy to get the novelty recognition from the reviewer for our proposed method. In addition to the discussed limitation in the global rebuttal, we will try to expand it further in the revision of the paper.
> > > > >
> > > > > Best regards,

---

### Author Rebuttal · Authors · 2023-08-07

First, we would like to thank the reviewers for their time and feedback. We would like to summarize some additional experiments in the rebuttal PDF and address common questions from the reviewers. Other questions are addressed in the corresponding rebuttal of reviewers.

1. **Additional experiments on deep generative modeling.** As suggested by Reviewer **QgRs** and Reviewer **krQV**, we conducted additional experiments on deep generative modeling using the framework proposed in [1]. We compared SW, vanilla DSW (which utilizes an MLP neural network to model the slicing distribution as suggested by Reviewer **krQV**), and IS-EBSW-e on CIFAR10 and CelebA datasets. For SW and IS-EBSW-e, we set L=100. For DSW, we set $L=10, T=10$, and reported the best result for $\lambda  \in \{1, 10, 50\}$ and slice learning rate ($\eta \in \{0.001, 0.01, 0.1\}$). We provided both quantitative results (FID score and IS score) and qualitative results (randomly generated images) in Figure 1 of the rebuttal PDF. Overall, we observed that IS-EBSW-e yielded the lowest FID score and the highest IS score. However, due to time limitations in the rebuttal, we recommend using this result as an example to showcase the versatility of EBSW, acknowledging that we ran experiments only once.

   [1] Amortized Projection Optimization for Sliced Wasserstein Generative Models, Nguyen et al.

2. **Comparison to DSW.** As suggested by Reviewer **krQV**, we also incorporated vanilla DSW into our applications. Firstly, we applied the gradient flow with DSW and presented the results in Figure 2 and Table 2 of the rebuttal PDF. We observed that vanilla DSW performed better than v-DSW but required more computation time. Overall, DSW still falls short compared to IS-EBSW-e in terms of performance and computational efficiency. Similarly, we included DSW in the deep point-cloud reconstruction application, where DSW's performance was comparable to v-DSW but inferior to our proposed IS-EBSW-e.

     Furthermore, we would like to emphasize the advantages of EBSW over DSW. Utilizing DSW involves two main steps: designing a family of distributions over the unit-hypersphere and selecting the best member through an interactive optimization procedure. Both steps present challenges; in the first step, selecting a suitable family remains an open question, and in the second step, optimization often suffers from local minima and is computationally expensive due to its iterative nature. Moreover, when performing statistical inference, using DSW (and other optimization-based SW, e.g., Max-SW) leads to a minimax problem known to be difficult to optimize. In contrast, EBSW is optimization-free, parameter-free, and stable. Although EBSW requires choosing an energy function, this process is easier and more flexible than selecting a family of distributions. Furthermore, EBSW can be computed exactly thanks to the asymptotic mixing property of importance sampling and Markov chains.

    In the paper, our criticism of DSW pertains to its computational form rather than its population form. Specifically, we provide an approximate budget constraint for DSW and EBSW (in terms of scaling constraint of $n \log n$) in our experiments, which means that DSW might not achieve its optimality. Additionally, we use the von Mises Fisher slicing distribution for DSW, which might not be the optimal choice. However, it's worth noting that the optimal choice for DSW is still unknown, and adopting a more expressive family could lead to increased memory and computation costs, which is unfavorable in a budget-constrained comparison setting.

    Overall, EBSW introduces a novel idea of energy-based slicing distribution that directly and explicitly conditions two input measures, resulting in a simple, effective, and efficient computation. Therefore, we believe it is easier to use EBSW in practice.

3.  **Increasing $p$ in SW.** As suggested by Reviewers **vaua**, we increased the value of p in SW, i.e., $p = 3, p = 10$ in gradient flow. However, we found that increasing $p$ does not lead to better performance for SW. The reason is that changing $p$ does not affect the slicing distribution in SW; it remains uniform. In contrast, EBSW adapts its slicing distribution changes adaptively. Therefore, increasing $p$ in SW does not have the same effect as in EBSW.


4.  **Limitations of the paper.** We will add a new paragraph to discuss the limitations of our paper, as suggested by Reviewers **QgRs** and **vaua**. In summary, the first limitation of EBSW is that its MCMC variants are not directly parallelizable, resulting in slow computation. Thus, conducting a more detailed investigation of customized MCMC methods for EBSW is crucial to unlocking its true performance. Additionally, the effective choice of energy-based functions is challenging. In the paper, we use simple and computationally effective energy-based functions, such as the exponential function and the polynomial function. However, there is ample room to explore other types of energy functions, such as the annealed exponential function and Lipschitz function, as suggested by Reviewer **vaua**. Finally, proving the triangle inequality of EBSW is challenging due to the expressiveness of the energy-based slicing distribution, which poses difficulties in deriving the two-sided sample complexity of EBSW, as questioned by Reviewer **krQV**. Overall, we believe these limitations pertain to the current paper itself, not EBSW.


Best regards,

Authors,

---

### Comment · Area_Chair_ncic · 2023-08-11
**Discussion period**

Dear reviewers and authors,

Thank you very much for your work on this submission and its evaluation. Now that the authors have responded to the reviews, I *strongly encourage* the reviewers to acknowledge the review, to look at other reviews and rebuttals for this submission, and to adjust their scores if needed. Thanks to those that have already done so.

Authors have the possibility to reply if further questions are needed, until the 16th.

Thank you very much to all,

Area Chair

---

> ### Author Response · Authors · 2023-08-15
> **Reminder on the end of discussion period**
>
> Dear reviewers and area chair,
>
> First, we would like to express our gratitude again for receiving constructive feedback and questions from the reviewers to improve our paper. We have answered all raised questions in our rebuttal and added new experiments as suggested. Therefore, we would be grateful if the reviewers could give us comments again. Moreover, we are happy to discuss new questions from the reviewers and the area chair. Finally, if the reviewers feel that there are no concerns left, we will be more than happy if the reviewers can increase the assessment score.
>
> Best regards,
>
> Authors

---

### Decision · Program_Chairs · 2023-09-21

**Decision:**

Accept (poster)

**Comment:**

There is an overall consensus among reviewers that this is an interesting submission that should be accepted, which is also my evaluation.